# SARS-CoV-2 nucleocapsid protein inhibits the PKR-mediated integrated stress response through RNA-binding domain N2b

**Chiara Aloise, Jelle G. Schipper, Arno van Vliet, Judith Oymans, Tim Donselaar, Daniel L. Hurdiss, Raoul J. de Groot** [ID]*, **Frank J. M. van Kuppeveld** [ID]*

Virology Section, Division of Infectious Diseases and Immunology, Department of Biomolecular Health Sciences, Faculty of Veterinary Medicine, Utrecht University, Utrecht, the Netherlands

* R.J.deGroot@uu.nl (RJdG); F.J.M.vanKuppeveld@uu.nl (FJMvK)

**Data Availability Statement:** The authors confirm that all data underlying the findings are fully available without restriction. All relevant data are

## Abstract

The nucleocapsid protein N of severe acute respiratory syndrome coronavirus 2 (SARS-CoV-2) enwraps and condenses the viral genome for packaging but is also an antagonist of the innate antiviral defense. It suppresses the integrated stress response (ISR), purportedly by interacting with stress granule (SG) assembly factors G3BP1 and 2, and inhibits type I interferon responses. To elucidate its mode of action, we systematically deleted and over-expressed distinct regions and domains. We show that N via domain N2b blocks PKR-mediated ISR activation, as measured by suppression of ISR-induced translational arrest and SG formation. N2b mutations that prevent dsRNA binding abrogate these activities also when introduced in the intact N protein. Substitutions reported to block post-translation modifications of N or its interaction with G3BP1/2 did not have a detectable additive effect. In an encephalomyocarditis virus-based infection model, N2b - but not a derivative defective in RNA binding—prevented PKR activation, inhibited β-interferon expression and promoted virus replication. Apparently, SARS-CoV-2 N inhibits innate immunity by sequestering dsRNA to prevent activation of PKR and RIG-I-like receptors. Similar observations were made for the N protein of human coronavirus 229E, suggesting that this may be a general trait conserved among members of other orthocoronavirus (sub)genera.

## Author summary

SARS-CoV-2 nucleocapsid protein N is an antagonist of innate immunity but how it averts virus detection by intracellular sensors remains subject to debate. We provide evidence that SARS-CoV-2 N, by sequestering dsRNA through domain N2b, prevents PKR-mediated activation of the integrated stress response as well as detection by RIG-I-like receptors and ensuing type I interferon expression. This function, conserved in human coronavirus 229E, is not affected by mutations that prevent posttranslational modifications, previously implicated in immune evasion, or that target its binding to stress granule scaffold proteins. Our findings further our understanding of how SARS-CoV-2 evades

within the paper and its Supporting Information files.

**Funding:** This work was supported by the European Union (Horizon 2020 Marie Skłodowska-Curie ETN "INITIATE", grant agreement number 813343) to FJMvK and by the Dutch Research Council NWO OCENW.KLEIN.344 to FJMvK. The funders had no role in study design, data collection and analysis, decision to publish, or preparation of the manuscript.

**Competing interests:** The authors have declared that no competing interests exist.

innate immunity, how this may drive viral evolution and why increased N expression may have been a selective advantage to SARS-CoV-2 variants of concern.

## Introduction

Vertebrate cells are provided with a diversity of interconnected sensors and effectors to timely detect and counter viral infection. Particularly dsRNA, an inevitable product of RNA and DNA virus replication, triggers a vigorous intracellular antiviral response [1]. For example, binding of dsRNA by 2'-5'oligoadenylate (2-5A) synthase (OAS) leads to enzyme activation and production of 2-5A, which in turn activates RNase L to stall the synthesis of viral proteins through non-specific RNA degradation [2–7]. Another dsRNA-activated pathway, the integrated stress response (ISR) with proteinase kinase R (PKR) as key sensor, entails inhibition of translation initiation and ultimately cell death. Upon dsRNA binding, PKR dimerizes, auto-phosphorylates and then proceeds to phosphorylate the alpha subunit of eukaryotic initiation factor eIF2, turning it into competitive inhibitor of guanine nucleotide exchange factor eIF2b. In consequence, the production of eIF2-GTP-(Met)tRNA$_i^{Met}$ ternary complex is downregulated, recognition of the initiation codon is blocked and cap-dependent translation-initiation prevented [8–11]. Polysome dissociation, ensuing activation of the stress response, results in an excess of stalled 48S preinitiation complexes, which are stored in cytoplasmic membrane-less organelles called stress granules (SGs) [12–14].

SGs are dynamic ribonucleoprotein assemblies that are formed through liquid-liquid phase separation with RNA binding protein Ras GTPase-activating protein-binding proteins 1 and 2 (G3BP1 and -2) functioning both as a molecular switches and main protein scaffolds together with T-cell-restricted intracellular antigen 1 (TIA1) and TIA1-related protein (TIAR) [15–19]. The SGs thus serve as deposits from which mRNAs, poised for translation through association with critical components of the translation machinery (40S ribosomal subunits, eIF4E, eIF4G, eIF4A, eIF4B, Poly(A) binding protein, eIF3, and eIF2), can be rapidly retrieved. SGs, however, are also considered a coordinating hub for the activation of other antiviral defense mechanisms like those of RIG-I-like receptors (RLRs). Indeed, RIG-I and MDA5, RLR-regulating PKR-activating protein (PACT), RLR-modulating ubiquitin ligases TRIM25 and TRAF2, and polyubiquitin chains are all recruited to ISR-induced SGs [20–23].

The antiviral mechanisms elicited by dsRNA are highly effective, even to such an extent that all known mammalian viruses code for one or more antagonists [11]. Coronaviruses, positive-stranded RNA viruses of exceptional genetic complexity, code for a universally conserved endonuclease (EndoU) that efficiently prevents simultaneous activation of host cell dsRNA sensors OAS, PKR and MDA5 through dsRNA decay [24–26]. Illustrating the importance of EndoU, mutants defective for EndoU are severely attenuated *in vivo*, and their replication in cultured primary macrophages is restricted presumably due to high basal expression levels of host sensors in these cells [25]. EndoU is derived by proteolytic processing of a large replicase polyprotein pp1ab, translated from the incoming viral genome, and essential to evade early innate and intrinsic antiviral host cell responses. Apparently, however, EndoU may not be sufficient to suppress dsRNA-mediated antiviral activities during later stages of the viral life cycle. To express the open reading frames (ORFs) downstream of the replicase gene, CoVs produce a 3' co-terminal nested set of subgenomic mRNAs from which the structural proteins are translated in addition to a variety of so-called accessory non-structural proteins. Some of the latter also have been shown to counteract dsRNA-mediated antiviral host responses. For example, the ns4a protein of Middle East Respiratory Syndrome coronavirus (MERS-CoV) prevents

PKR-mediated stress by sequestering dsRNA [27,28], whereas the MERS-CoV ns4b protein is a phosphodiesterase (PDE) and antagonizes the OAS-RNase L pathway by enzymatically degrading 2-5A activators [29]. Non-related PDEs, ns2 proteins, are found in members of the subgenus *Embecovirus*, including human coronavirus OC43 [30]. Finally, a specific inhibitor of the ISR was found in gammacoronaviruses of cetaceans. The beluga whale coronavirus accessory protein 10 (BWCoV acP10) blocks p-eIF2-eIF2B association to allow continued formation of the ternary complex and unabated global translation even at high p-eIF2 levels that would otherwise cause translational arrest [31].

Apparently, CoVs rely on redundancy in antagonists and antagonistic mechanisms to effectively counter dsRNA-induced antiviral host responses. Likewise, the current pandemic severe acute respiratory syndrome coronavirus 2 (SARS-CoV-2) codes for an ISR inhibitor additional to EndoU [32]. Its nucleocapsid protein (N) was reported to block the ISR and to inhibit SG formation in a PKR- and G3BP1-dependent fashion [33–36]. In apparent concordance, proteomics studies and structural analyses provided evidence for physical interactions between N and the SG key components G3BP1 and G3BP2 [33,37–40].

Coronavirus N proteins have a modular structure with N-terminal and C-terminal RNA-binding domains, called N1b and N2b respectively, bounded by largely disordered regions. Here, we took a reductionistic approach involving systematic deletion and individual transient expression of the different regions and domains. We show that the N2b domain is critical and sufficient to counter the ISR. Single site mutations in N2b that block dsRNA binding prevent PKR activation. This activity is not affected by posttranslational modifications elsewhere in the N protein that are known to regulate RNA binding nor dependent on physical interaction with G3BP1 and G2BP2. Using the encephalomyocarditis virus as a surrogate infection system we show that N2b domain prevents PKR-mediated activation of the ISR and suppresses IFNβ expression also in virus-infected cells. Our findings suggest that in addition to a function in replication and genome packaging, SARS-CoV-2 N functions as a classical antagonist of dsRNA-induced host defense.

## Results

### SARS-CoV-2 N inhibits PKR-induced ISR

Transfection of cells with specific expression plasmids like pEGFP-N3 triggers the ISR through dsRNA-mediated PKR activation, resulting in translation arrest and the formation of SGs (**Fig 1A**) [27,41–43]. The dsRNA arises from spurious bidirectional transcription of plasmid sequences [44] and can be readily detected in transfected cells with dsRNA-specific antibodies [27]. This phenomenon allows for a convenient method to identify potential viral ISR antagonists by transiently expressing these proteins genetically fused to enhanced green fluorescent protein (EGFP) [27,31]. The expression levels of these fusion proteins, as judged by fluorescence microscopy, can then be compared to that of EGFP alone as an indicator for translation arrest and the prevention thereof. In addition, SG formation can be assessed by immunofluorescence analysis (IFA) by staining the transfected cells for established SG markers like G3BP1, G3BP2, eIF3 and TIA-1 (**S1A Fig**). This procedure previously allowed us to identify several stress antagonists including MERS-CoV 4a and Beluga whale coronavirus AcP10 [27,31].

SARS-CoV-2-infected Vero E6-TMPRRS2 cells, identified by detection of dsRNA, were virtually devoid of SGs, suggestive of virus-induced suppression of the ISR (**S2 Fig**). While probing SARS-CoV-2 proteins for a potential role in ISR inhibition, we noted that in transfected wildtype (wt) HeLa cells, the expression levels of the SARS-CoV-2 N-EGFP fusion protein were strongly increased as compared to the EGFP control. This pattern of enhanced expression in a sizeable population of transfected cells was similar to that observed for established ISR

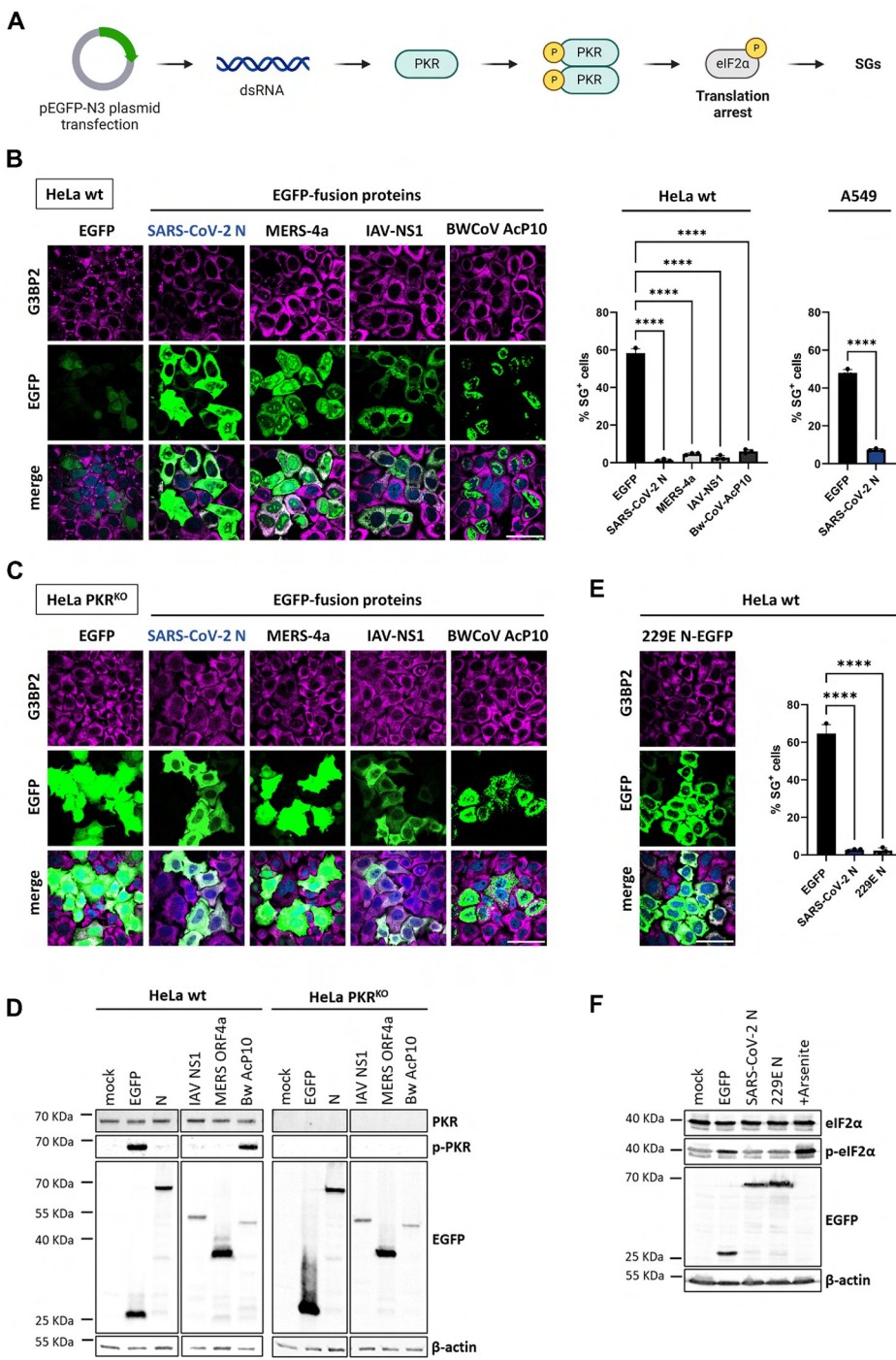

**Fig 1. SARS-CoV-2 N inhibits PKR-induced ISR, preventing translation arrest and SG formation in HeLa and A549 cells.** (*A*) Schematic representation of the PKR-induced ISR activation upon the pEGFP-N3 transfection in eukaryotic cells. Created with BioRender.com. (*B*) HeLa and A549 cells were transfected with expression vector pEGFP-N3 (EGFP) or pEGFP-N3 derivatives encoding SARS-CoV-2 N, MERS-4a, IAV-NS1 or BwCoV-AcP10 genetically fused to EGFP. Induction of the ISR via dsRNA-mediated PKR activation or suppression thereof was assessed by comparing EGFP fluorescence intensity and SG formation as detected by immunofluorescence staining for G3BP2 (see also **S1A Fig**). (*left panel*) Transfected HeLa cells; (immune)fluorescence microscopy images, representative results. Scale bars: 50 μm. Settings of image acquisition (laser intensity, exposure time) and processing conditions used throughout were chosen to avoid over-exposure in high-expressing cells. Note that under these conditions, EGFP expression is too low to be detected in a sizeable population of stressed and non-stressed cells (see

also **S1B Fig**). (*right panel*) Bar graphs for Hela and A549 cells showing the percentage of EGFP expressing cells containing G3BP2-positive SGs. The *results* are *representative* of three independent experiments counting >200 cells per sample. Standard deviation indicated by error bars; *** p = 0.001, **** p< 0.001, ns = not significant (one-way ANOVA with Dunnett post hoc test). (**C**) HeLa PKR$^{KO}$ cells transfected as in B. Representative (immuno)fluorescence microscopy images are shown. (**D**) Western-blot analysis for PKR, phosphorylated PKR (p-PKR), EGFP-fused proteins and β-actin. HeLa wt and PKR$^{KO}$ cells, mock-transfected or transfected with indicated plasmid were lysed at 24 hr. Of note, the larger yield of EGFP versus N-EGFP in either cell type is counter intuitive but can be explained from the fact that (i) this is an ensemble measurement (for the total transfected cell population) versus the analysis of individual cells by fluorescence microscopy and (ii) basal expression levels of the codon optimized EGFP *prior* to ISR activation will exceed those the N-EGFP fusion protein, which is non-codon optimized and three times larger in size (see also **S4 Fig**). (**E**) Inhibition of SG formation and translational arrest by pEGFP-N3-expressed N proteins of MERS-CoV and HCoV-229E. (Immuno)fluorescence analysis (*left panel*), quantitative representation of the results and statistical analysis as in B. (**F**) Western-blot analysis for eIF2α, phosphorylated eIF2α (p-eIF2α), EGFP-fused proteins and β-actin. HeLa cells were transfected to express EGFP or EGFP-tagged N proteins of SARS-CoV-2 and HCoV 229E. Mock-transfected cells were either left untreated (mock) or treated with sodium arsenite (+Arsenite) to induce eIF2α phosphorylation.

antagonists (**Fig 1B**). Moreover, like these antagonists, SARS-CoV-2 N significantly reduced the number of transfected cells with stress granules both in wildtype HeLa (HeLa wt) as well as in A549 cultures (**Fig 1B**). SG suppression was observed also in cells with fluorescence intensities just above the background, suggesting that low levels of N-EGFP are already sufficient to prevent SGs from forming.

To confirm that expression of EGFP was suppressed due to PKR-induced ISR-mediated translational arrest, we repeated the experiments in PKR-deficient HeLa PKR$^{KO}$ cells. Indeed, EGFP expression was increased and SGs were absent in these cells (**Fig 1C**). In further accordance, PKR was activated -as indicated by PKR phosphorylation- in HeLa wt cells expressing EGFP but not in cells expressing SARS-CoV-2 N-EGFP. (**Fig 1D**, left panel). Moreover, phosphorylation of eIF2α, as observed in pEGFP-transfected cells, was prevented by SARS-CoV-2 N-EGFP. (**Figs 1F and S3**). Finally, Western blot analysis confirmed that N-EGFP was expressed to similar levels in HeLa wt and PKR$^{KO}$ cells. Levels of EGFP, however, were strongly increased (about 7-fold) in HeLa PKR$^{KO}$ cells (**Fig 1D**, right panel; **S4 Fig**). The combined findings identify SARS-CoV-2 N as an ISR antagonist that prevents translational arrest and ensuing SG formation by acting as a PKR inhibitor.

The N protein of another betacoronavirus, MERS-CoV, was previously noted to suppress SG formation too [35]. Interestingly, we observed inhibition of eIF2α phosphorylation, translational arrest and SG formation also in cells expressing the N protein of human alphacoronavirus 229E (HCoV-229E) (**Figs 1E**, **1F and S3**). Apparently, N-mediated suppression of the ISR is not unique to betacoronaviruses but may well be a general trait conserved among members of other orthocoronavirus (sub)genera.

## Domain N2b is essential and sufficient for suppression of PKR-induced SG formation

The N proteins from the four CoV genera share only 27 to 30% sequence identity but are conserved in their modular organization (**Fig 2A**) ([45,46] for a review see [47]). They are comprised of two ordered domains, the N-terminal domain (N1b also called NTD) and the C-terminal domain (N2b aka CTD) (in SARS-CoV-2 N, residues 49–175 and 248–365, respectively). N1b and N2b are separated and flanked by segments, predicted to be at least partially disordered (N1a, N2a and N3). N1b and N2b have been implicated in RNA binding and, in case of N2b, also in N dimerization. Indeed, in infected cells and upon heterologous expression, CoV N proteins form homodimers that, in turn, assemble into tetramers mediated by N3 [48]. The N2a central spacer contains a serine- and arginine-rich (SR) region, immediately downstream of N1b, the regulated phosphorylation of which is deemed important for N

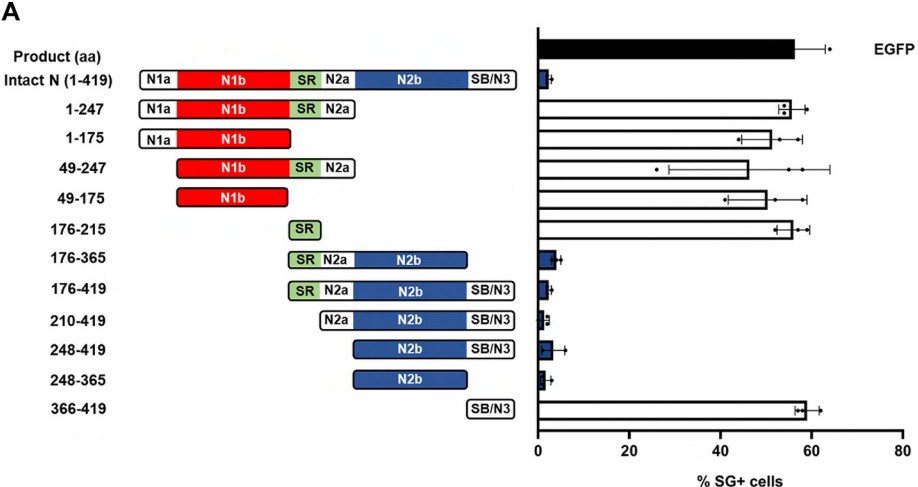

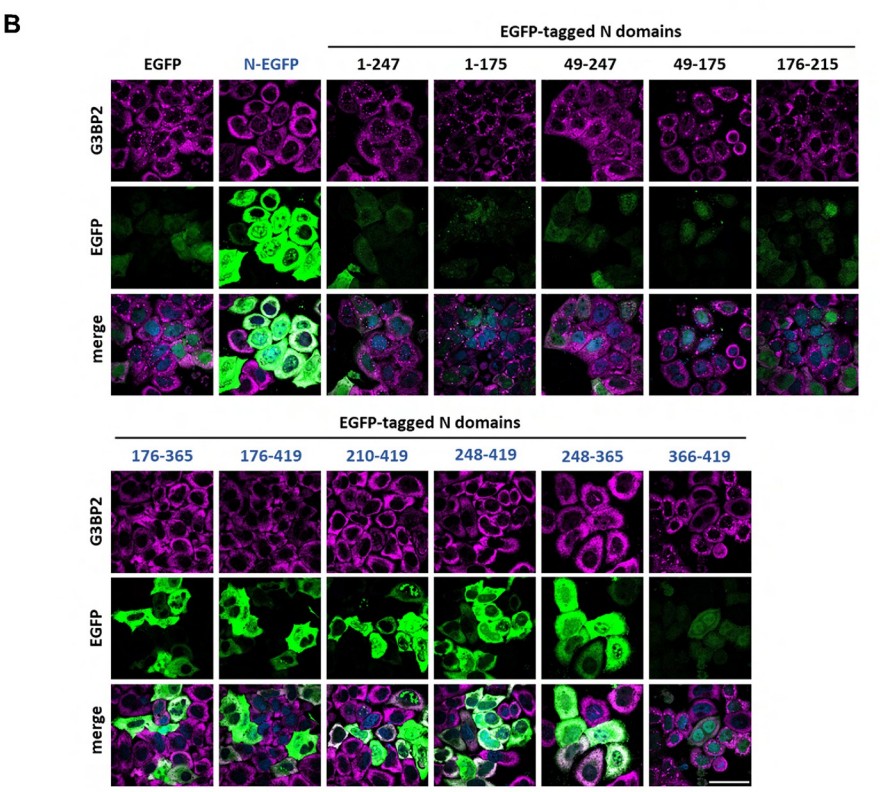

**Fig 2. Nucleocapsid domain N2b suppresses translational arrest and SG formation.** (*A*) Schematic representation of truncated derivatives of SARS-CoV-2 N protein fused to EGFP (*left panel*). The proteins were overexpressed from pEGFP-N3-based vectors in HeLa cells and percentages of SG-positive cells were determined as in Fig 1A (*right panel*). The data are representative of three independent experiments with more than 200 cells counted for the presence of G3BP2-positive SGs per individual sample. Standard deviation indicated by error bars. For a statistical analysis of the results, see **S1 Table**. (*B*) Representative results from (immuno)fluorescence analysis of HeLa cells transfected to express the SARS-CoV-2 N protein derivatives. Scale bar: 50 μm.

function during different stages of the CoV replication cycle [49–51]. The primary function of N is in virus assembly. It condenses newly produced gRNA into helical nucleocapsids, apparently with N2b controlling target selectivity [52,53], and then drives envelopment by binding to the viral membrane protein M via N3. However, N has several functions auxiliary to virion morphogenesis. At the very start of the infectious cycle, N is essential for the initiation of infection by the incoming gRNA through association of SR with the cytosol-exposed ubiquitin1 domain of replication organelle pore protein nsP3 [51,54]. Moreover, purportedly relevant to ISR suppression, N binds to G3BP1 and 2 through segment N1a [38,40,55].

To determine the molecular basis for ISR suppression, we constructed a library of N mutants with partially disordered regions and domains either systematically deleted or expressed in isolation. As shown in **Fig 2**, suppression was lost upon deletion of subdomain N2b. Moreover, expression of N2b in isolation caused a reduction in SGs to an extent similar to that of full-length N. The data therefore suggested that ISR inhibition and SG suppression as observed for the intact N protein is mediated at least in part by N2b.

## N2b mutations that disrupt dsRNA binding abrogate suppression of SG formation

N2b binds both single and double stranded oligonucleotides, whether RNA or DNA [47,56,57]. Hence, a possible mechanism for N2b to prevent PKR-induced activation of the ISR is by sequestering dsRNA. Crystal structural analysis revealed that N2b homodimers form a rectangular slab with wide faces of 45 Å × 35 Å in dimension [56]. One is comprised of an interlaced inter-molecular four-stranded β-sheet and predominantly negatively charged, the other of two α-helical regions separated by a shallow central, positively charged groove thought to be the oligonucleotide binding site [47,56,58,59]. Within the groove, there are several conserved positively charged residues with their surface-exposed side chains seemingly poised for interaction with nucleic acid, particularly $Lys^{257}$ and $Lys^{261}$ (**Fig 3A**). The binding by bacterially-expressed N2b of synthetic RNA oligonucleotides, whether single or double-stranded, SARS-CoV-2 derived (highly conserved 3' UTR stem-loop II motif, s2m; [60]) or a scrambled version thereof, can be readily demonstrated by electrophoretic mobility assay (EMSA) (**Fig 3B**).

To test whether $Lys^{257}$ and $Lys^{261}$ are involved in RNA binding and whether such binding is important to counteract the ISR and suppress SG formation, they were replaced by Ala, separately and in combination. Indeed, dsRNA binding was either significantly reduced or lost beyond detection upon substitution of $Lys^{257}$ and $Lys^{261}$, respectively (**Fig 3B**). In accordance, upon Ala substitution of $Lys^{261}$ or $Lys^{257}$, the mutant N2b proteins no longer rescued protein synthesis (**Fig 3C**) and lost their capacity to suppress the formation of SGs, either completely ($Lys^{261}$) or partially ($Lys^{257}$) (**Fig 3D**). Notably, Ala substitution of other surface-exposed charged residues ($Gln^{272}$, $Gln^{289}$, $Arg^{276}$, $Arg^{293}$), not located within the RNA binding groove but rather implicated in N2b inter-dimer association [47,57], did not detectably affect SG formation (**Fig 3D**).

Importantly, also in the intact N protein, $Lys^{261}$Ala substitution abrogated inhibition of translational arrest as well as SG formation and so did the $Lys^{257}$Ala mutation (**Fig 3C and 3D**). It is unknown why the latter mutation exerts a stronger effect in the context of the full-length protein than in N2b. The combined data, however, do show that $Lys^{261}$ and $Lys^{257}$ are required for nucleic acid binding by N2b and suggest that this capacity to bind RNA is essential for SG suppression. Moreover, the observation that also for the intact N protein SG suppression was reduced to background levels by single site $Lys^{261}$Ala and $Lys^{257}$Ala substitutions suggests that inhibition of PKR-induced ISR by SARS-CoV-2 N critically relies on N2b and

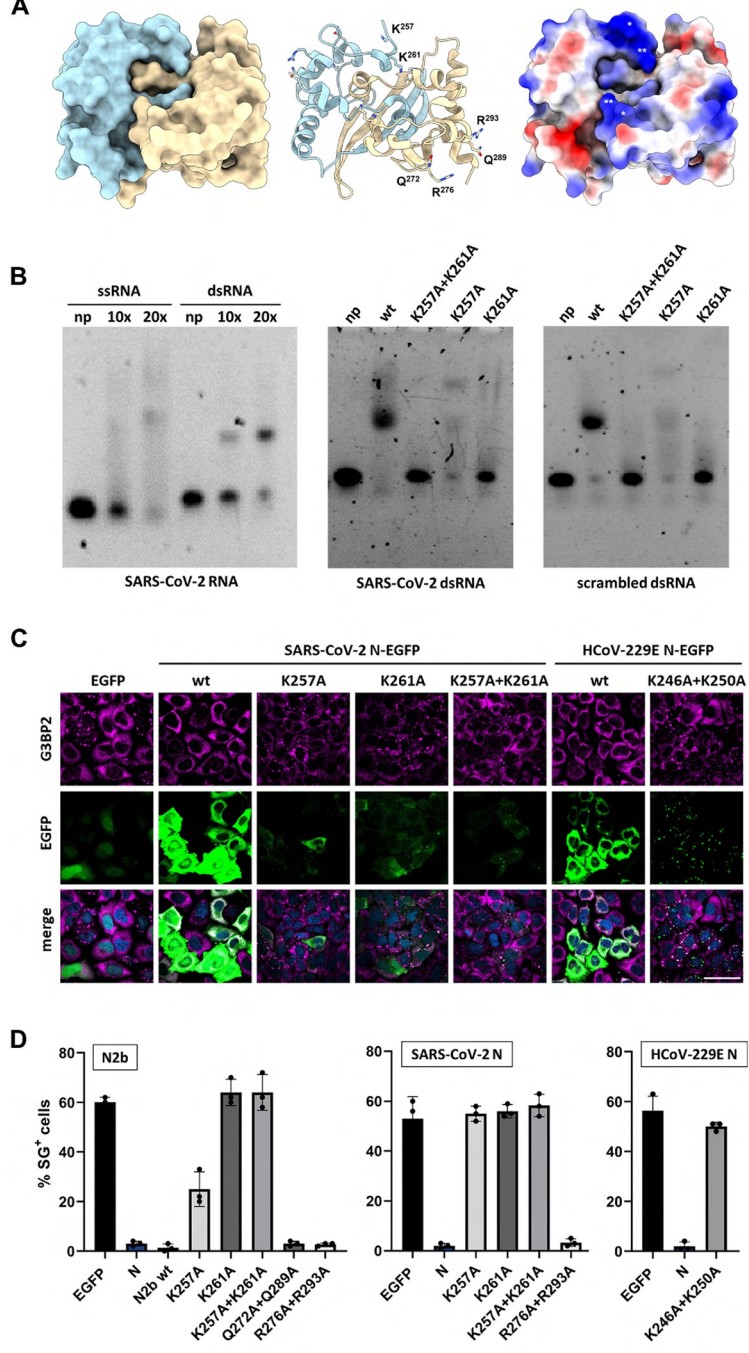

**Fig 3. N2b mutations that disrupt dsRNA binding abrogate suppression of translational arrest and SG formation.** (*A*) (*left panel*) Surface representation and (*middle panel*) cartoon representation of the SARS-CoV-2 N2b dimer (PDB ID code 7C22 [59] with monomers in light blue (chain A) and wheat (chain B). Side chains of mutated charged/polar residues labeled and shown in sticks (Chain A: Lys$^{257}$ and Lys$^{261}$; Chain B: Q$^{272}$, Q$^{289}$, R$^{276}$ and R$^{293}$). (*right panel*) N2b dimer surface representation colored according to calculated charge distributions, displaying the positively-charged central RNA-binding groove. Critical residues Lys$^{257}$ (single asterisk) and Lys $^{261}$ (double asterisk) marked for both monomers. Top view images at a forward 45° tilt; generated with UCSF ChimeraX version 1.6.1 [79]. (*B*) Electrophoretic mobility shift assays (EMSA). (*left panel*) EMSA with bacterially expressed N2b domain and single (ss) and double-stranded (ds) RNA oligonucleotides, designed after SARS-CoV-2 stem-loop II motif (s2m). Assays were performed with N2b in 10- or 20-fold molar excess. Non-bound RNA was included as a control (np, no protein). (*middle panel*) The effect of N2b amino acid substitutions on binding of s2m dsRNA or (*right panel*) a scrambled

version thereof (right). EMSAs performed with N2b and mutant derivatives in 20-fold molar excess (*middle and right panels*). (*C-D*) Mutational analysis of SARS-CoV-2 N2b, full-length SARS-CoV-2 N and full-length HCoV-229E N. Select surface-exposed charged residues were substituted by Ala either individually or in combination as indicated and the effect on IRS-induced translational arrest (C) and SG formation (D) was analyzed in HeLa cells as in **Fig 1**. For a statistical analysis of the results, see **S2 Table**.

N2b-mediated RNA binding. Notably, the N protein of alphacoronavirus HCoV-229E also lost its capacity to block SG formation when the orthologous N2b residues, Lys[246] and Lys[250], were substituted (**Fig 3C and 3D**), suggestive of a common mechanism for ISR inhibition through the binding of RNA.

## N2b-mediated inhibition of PKR-induced SG formation is not affected by posttranslational modifications of N

Previous studies identified N as a multifunctional protein involved in different aspects of the viral replication cycle beyond viral assembly. Its activity apparently is regulated by posttranscriptional modifications such as differential phosphorylation of the SR domain, which alters RNA binding affinity [61,62], and acetylation at Lys[375], reportedly essential for liquid–liquid phase separation of N-RNA ribonucleoprotein complexes [63]. Arginine methylation of SARS-CoV-2 nucleocapsid protein at Arg[95] and Arg[177] was reported to regulate RNA binding and its ability to suppress stress granule formation [33].

To test the importance of these posttranscriptional modifications on N2b-mediated inhibition of SG formation, we performed site-directed mutagenesis in the context of the intact N-EGFP fusion protein. Suppression of SG formation was not affected by Ala substitution of Arg[95] or Lys[375] indicating that methylation and acetylation of N is not essential (**Fig 4A and 4B**).

An N-EGFP derivative defective in phosphorylation of the SR segment was constructed by replacing 13 out of the 14 Ser residues by Ala. The resulting mutant, N-(13S>A)-EGFP, displayed a cellular distribution distinctively different from that of the parental wildtype protein. Instead of an even distribution throughout the cytoplasm, the mutant protein clustered in what looked like large aggregates. These local deposits were enriched for G3BP2 but mostly devoid of eIF3 (**Figs 4A and S5**) and differed in size and appearance from typical SGs. Importantly, however, the expression levels of the N-13S>A mutant were like those of wildtype N, as judged by EGFP fluorescence intensity observed in IFA, and consistently higher than those of EGFP alone (**Fig 4A**). We interpret the findings to indicate that the SR mutations may cause the N protein to aggregate but they do not affect inhibition of ISR-induced translational arrest. The results confirm those of our systematic deletion analysis (**Fig 2**).

## N2b-mediated inhibition of SG formation is not affected by disruption of the G3BP binding motif

A ΦxFG motif within N1a (residues 15–18), required for association with G3BP1 and 2 [40], was recently proposed to rewire the G3BP interactome to disrupt stress granules [38]. To investigate a possible role of N-G3BP interaction in suppressing PKR-induced SGs, we mutated the N1a ΦxFG motif through Ala substitution of key residue Phe[17] (mut 1A) or by a combination of Ala substitions: Arg[14] and Ile[15] (mut 2A), Ile[15], Phe[17] and Gly[18] (mut 3A), or Ile[15], Thr[16], Phe[17] and Gly[18] (mut 4A). In each case, binding to endogenous G3BP1 was no longer detectable by pull down assay (**Fig 4C**) yet suppression of SG formation was not affected (**Figs 4A**, **4B and S6**). Conversely, N proteins with N2b mutations to abrogate dsRNA binding no longer blocked the ISR (**Figs 3 and 4B and 4C**) nor inhibited SG formation, even though N1a was still intact to bind G3BP1 to a similar extent as wildtype N (**Fig 4C**; for

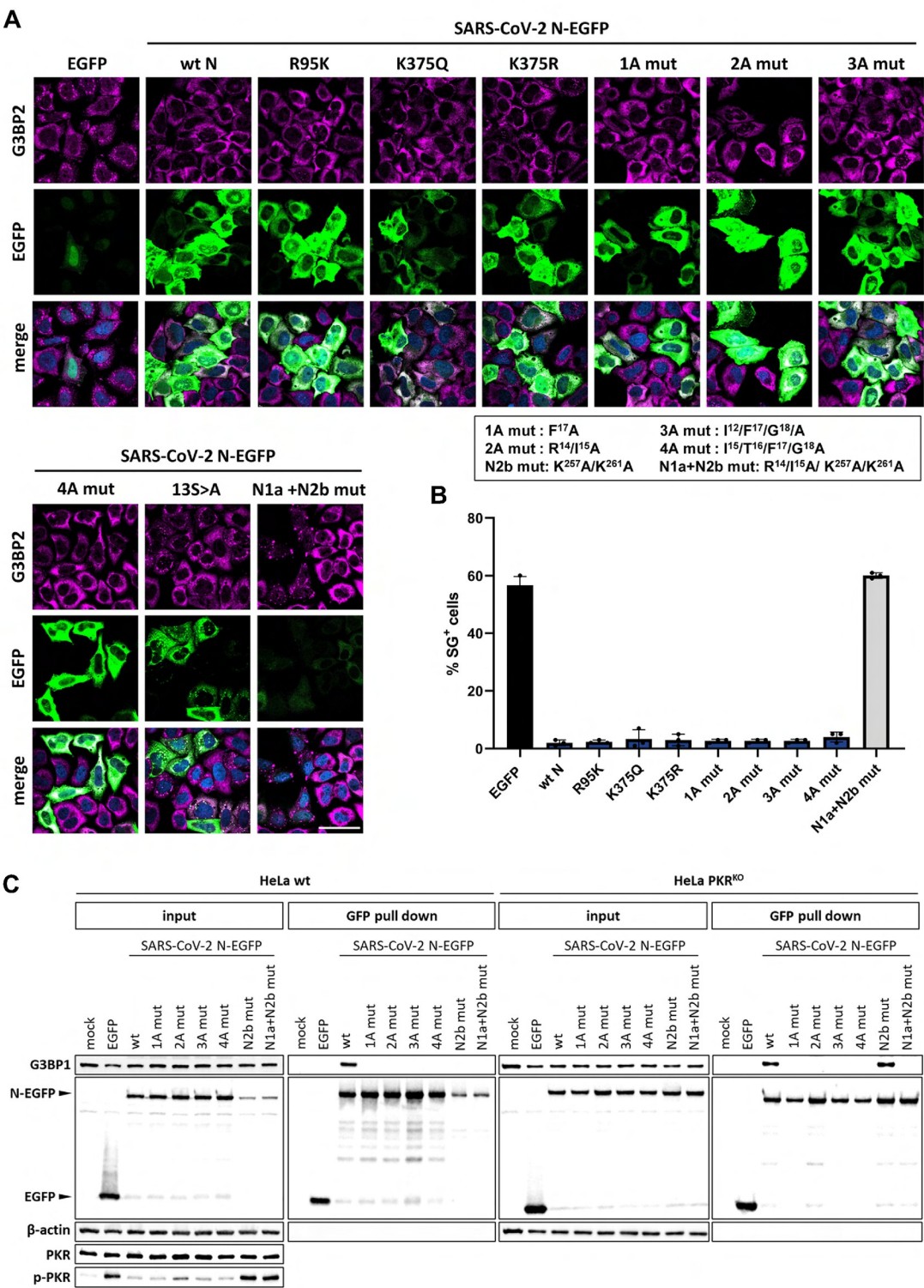

**Fig 4. N-mediated suppression of SG formation and ISR-induced translational arrest is not affected by posttranslational modifications or G3BP1 interaction.** (*A*) The effect on N-mediated ISR inhibition by mutations introduced to prevent posttranslational modifications ($K^{375}Q$, $K^{375}R$, $R^{95}K$), to disrupt the G3BP1/2 binding site (1A-4A mutants) or to prevent phosphorylation of the SR element (13S>A) was tested in HeLa cells by (immune)fluorescence analysis as in **Fig 1**. Scale bar: 50 μm. (*B*). Quantification of the results in (A) representative of three independent experiments, with >200 cells counted per

sample as in Fig 1. (*C*) Pull down assay to test SARS-CoV-2 N mutants for their association with endogenous G3BP1 in HeLa wt (left panel) and PKR[KO] cells (right panel).

supportive IFA evidence see **S6 Fig**). Note that for these N2b mutants N-G3BP association could not readily be assessed in HeLa wt cells by Western blot analysis. As a result of PKR-activation (**Fig 4C**), eIF2α phosphorylation (**S3 Fig**) and ensuing ISR-mediated translational arrest (**Figs 4A and S6**), the expression levels of these mutants were strongly reduced as compared to those of wt N and the amounts of coprecipitated G3BP1 were below the detection level (**Fig 4C**, 'HeLa wt'). Unperturbed G3BP binding, however, was evident, when assessed in HeLa PKR[KO] cells (lane marked 'N2b mut') and lost upon concomitant mutation of the N1a ΦxFG G3BP binding motif (lane marked 'N1a+N2b mut') (**Fig 4C**). The findings indicate that SARS-CoV-2 N-G3BP interaction is not required for inhibition of the ISR, at least not upon induction of the ISR via the PKR signaling pathway under the conditions applied. Moreover, this interaction is also not sufficient to prevent PKR-induced formation of SGs or to promote their disassembly.

To study whether G3BP-binding might still affect SG formation, HeLa PKR[KO] cells were transfected to express wildtype N or mutant derivatives and subjected to arsenite/heme-regulated inhibitor kinase (HRI)-induced stress. This approach allowed us to study the importance of N-G3BP1 interaction more directly, i.e. without transfection induced PKR activation and N-mediated inhibition of PKR as complicating factors. Under these conditions, wt N-EGFP still inhibited SG formation as compared to EGFP alone albeit to a modest extent (**Fig 5**). This phenomenon may be ascribed to G3BP sequestration, because the observed reduction in the number of SG-producing cells was largely abrogated by mutations in the N1a ΦXFG G3BP binding motif (**Fig 5B**). SG inhibition seemed dependent on N-EGFP expression levels. The effect was more pronounced in cells with high EGFP fluorescence intensity (**Fig 5B** and **5C** and **S3 Table**). Conversely, N2b-dependent loss of PKR-induced SGs in wildtype HeLa cells was already observed at very low levels of N (**Fig 1B**).

## Suppression of PKR-induced ISR by coronavirus N proteins is mediated predominantly by N2b

To corroborate our observations, we measured N-mediated rescue of ISR-induced translational arrest also by flow cytometry. To this end, we used a cotransfection assay with red fluorescence protein (RFP) expressed from vector pcDNA-RFP serving as reporter [27]. Cotransfection with ISR-inducing plasmid pEGFP-N3 strongly inhibited RFP production (**Fig 6**, top; see also **S7 Fig**). RFP expression was rescued in cells co-expressing N2b but not by its RNA-binding deficient derivative N2b-K[257]A/K[261]A. RFP expression was also rescued by the intact N proteins of either SARS-CoV-2 or HCoV-229E and to similar levels by SARS-CoV-2 N mutants defective for G3BP1/2 binding (N-R[14]A/I[15]A) or N1b methylation (N-R[95]K) (**Fig 6**). In contrast, inactivation of N2b through the K[257]A/K[261]A double mutation in the intact N protein of either SARS-CoV-2 or HCoV-229E significantly reduced RFP expression (**Fig 6**). Intriguingly, however, RFP expression levels were still higher than those observed for EGFP (p-value 0.0014) or N2b-K[257]A/K[261]A (p-value 0.0195). The data suggest that suppression of the PKR-induced ISR by coronavirus N proteins is mediated predominantly by the N2b domain though not exclusively. Apparently, N counteracts translational arrest also through other domains via alternative mechanisms, but this contribution only becomes detectable upon N2b inactivation. Saliently, however, disruption of N-G3BP interactions through substitutions in segment N1a did not have an additive effect when tested in combination with the N2b mutations (**Fig 6**; panel 'SARS2 N N1a+N2b mut').

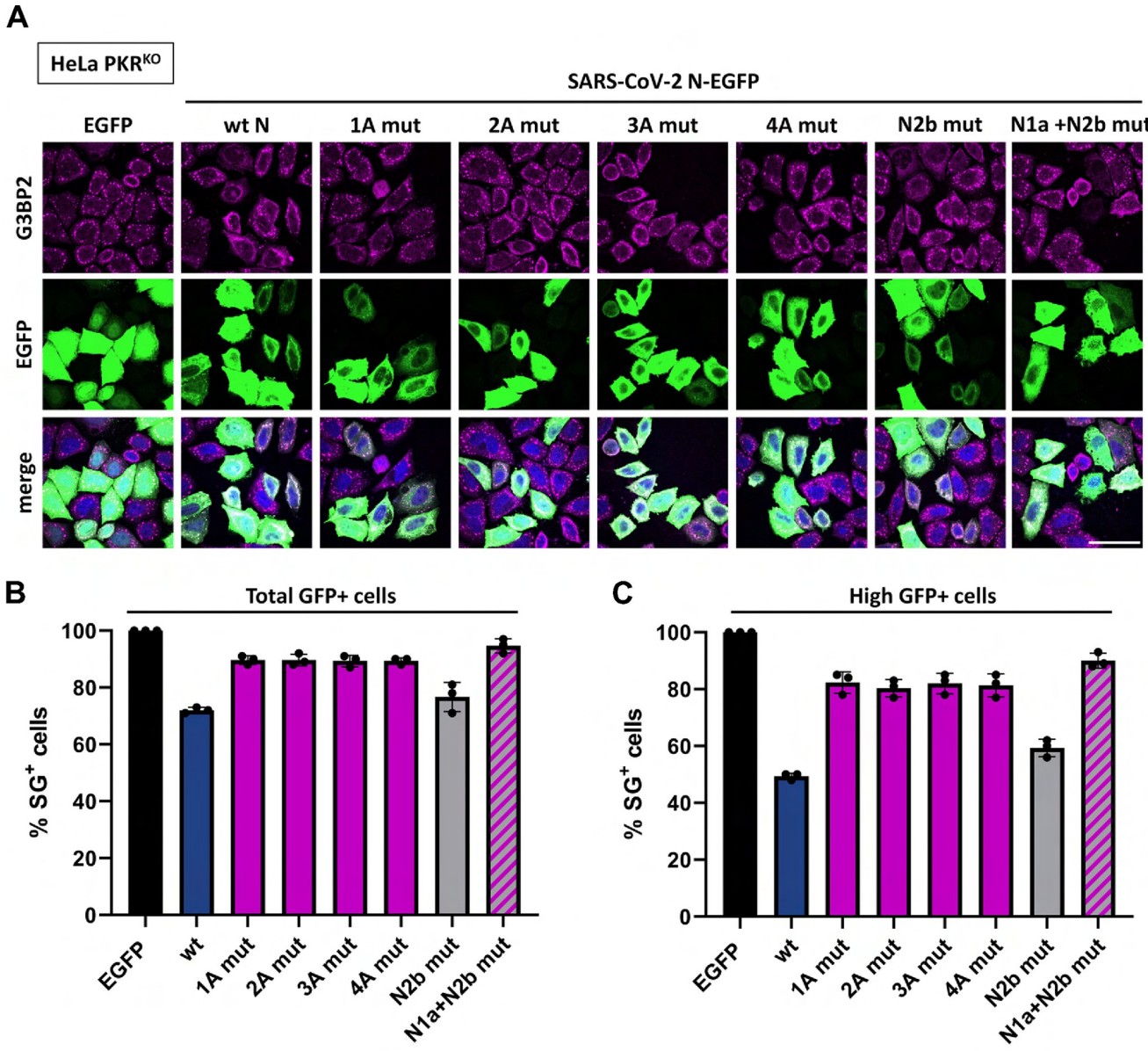

**Fig 5. The effect of N-G3BP interaction on arsenite-induced SGs in HeLa PKR$^{KO}$ cells.** HeLa PKR$^{KO}$ cells were transfected to express EGFP, SARS-CoV-2 N wt, N mutants defective in G3BP binding (1A-4A mut), N-K$^{257}$A+K$^{261}$A (N2b mut) or a mutant 'N1a + N2b mut' with substitutions in both the G3BP binding site (2A mut) and N2b (K$^{257}$A+K$^{261}$A). At 24 hrs post transfection, cells were sodium arsenite-treated to induce HRI-mediated ISR with ensuing formation of SGs. (**A**) Representative (immune)fluorescence images. Scale bar: 50 μm. (**B**) Quantification of the results based on three independent experiments by counting all cells with detectable EGFP fluorescence or (**C**) highly expressing cells exclusively. For the statistical analysis, see **S3 Table**.

## N2b-mediated suppression of the ISR and type I interferon response in virus-infected cells

Whereas the results provide conclusive evidence for ISR suppression upon transient expression of N in transfected cells, the question arises whether this phenomenon also occurs during natural infection and whether it is relevant for evasion of the innate host immune response. Unfortunately, the essential functions of N at multiple stages of the coronavirus life cycle, both during early replication as well was in genome packaging during virion assembly, precludes a

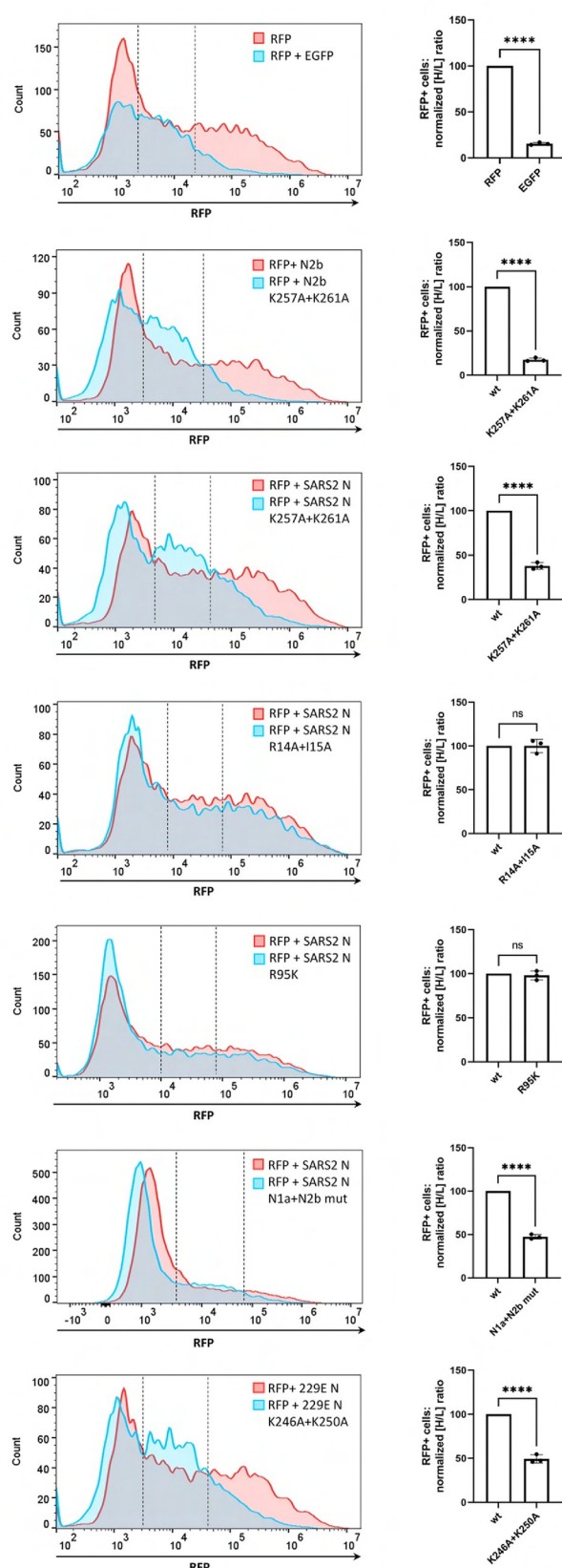

**Fig 6. Quantitative assessment of N-mediated rescue of ISR-induced translational arrest.** HeLa cells were transfected to express EGFP, SARS-CoV-2 N2b, the full-length N proteins of SARS-CoV-2 and HCoV-229E, and mutants thereof from pEGFP-N3-based expression vectors to induce PKR-activated ISR. The capacity of these proteins to rescue translational repression of red fluorescent protein (RFP) *in trans* or lack thereof was measured by flow cytometry at 24 hr post transfection (*left panels*) and fluorescence microscopy (see **S7 Fig**). Representative flow cytometry histograms shown. The transfected cells were divided into non-RFP-expressing, low (L) RFP-expressing and high (H) RFP-expressing populations (see dashed lines in histograms). For each mutant protein, the [H/L] ratio was calculated and normalized, with those of the corresponding wildtype proteins set at 100%. The bar graphs show mean values with standard deviations based on three independent experiments (unpaired t-test; ****, P<0.001; ns, non-significant) (right).

straightforward reverse genetics approach. Thus, the construction of recombinant SARS-CoV-2 mutants with an N protein defective in N2b RNA binding was deemed a nonviable option. We therefore resorted to a well-established alternative model based on recombinant encephalomyocarditis virus mutant EMCV-L$^{zn}$, in which the autologous ISR antagonist (the leader protein L) is inactivated. The mutant virus is no longer able to counteract the ISR [64–66] and hence provides a convenient platform to identify and characterize ISR antagonists of other viruses.

Our initial experiments were performed with an EMCV mutant that coded for a chimeric polyprotein with N2b at its N-terminus, downstream of the first five residues of EMCV L$^{zn}$ that are important for efficient IRES-mediated translation [67] and two additional residues encoded by an engineered *Xho*I site. This virus, however, failed to suppress the ISR. We noted that in the fusion protein, two negatively charged residues, E$^6$ (from L) and E$^8$ (encoded by the *Xho*I sequence) are immediately upstream of N-terminal N2b residues K$^{248}$ and K$^{249}$ (residues 9 and 10 of the chimera) and proximal to critical N2b residues K$^{257}$ and K$^{261}$ **S8 Fig**). Arguing that this might affect N2b RNA binding, we introduced into the EMCV-L$^{zn}$ genome an N-terminally extended N2b with the N2a region serving as a spacer (**Figs 7A** and **S8**). Indeed, in cells, infected with the resulting virus EMCV-L$^{zn}$-N2aN2b$^{wt}$, SG formation was suppressed to 30% of that caused by EMCV-Lzn, i.e. to levels similar to those reported for EMCV derivatives with L replaced by established PKR inhibitor MERS-CoV ns4a (**Fig 7B and 7C**). In accordance, in cells infected with the N2aN2b$^{wt}$ virus, levels of phosphorylated PKR were consistently low, comparable to those in wildtype EMCV-infected cells and, as calculated from band densities, 4.5 to 6-fold lower than those in cells infected with EMCV-L$^{zn-}$. Thus, in its activity N2aN2b$^{wt}$ mimicked MERS-CoV ns4a which prevents PKR activation by sequestering dsRNA [27] but differed from AcP10 which inhibits the ISR at a level downstream of PKR [31]. Like MERS-CoV ns4a and AcP10, N2aN2b restored replication efficacy of EMCV-L$^{zn}$ to near wild-type levels as based on expression of the viral capsid proteins (**Fig 7D**).

To test whether N2aN2b, like ns4a, also suppresses the type I interferon response, we tested for phosphorylation of IRF3 by Western blot analysis and measured the levels of IFN-β mRNA by qRT-PCR. Indeed, in cells infected with EMCV-L$^{zn}$-N2aN2b$^{wt}$ IRF3 phosphorylation was inhibited (**Fig 7D**) and IFN-β expression was reduced as compared to EMCV-L$^{zn}$ infected cells to levels observed in cells infected with EMCV wt or EMCV-L$^{zn}$-MERS-ns4a (**Fig 7E**). In contrast, in cells infected with EMCV-L$^{zn}$-N2aN2b-K$^{257}$A+K$^{261}$A, encoding an inactive N2b, there was no detectable suppression of SG formation, PKR and IRF3 were phosphorylated, and viral replication was delayed to a similar if not larger extent than in EMCV-L$^{zn}$-infected cells (**Fig 7B and 7D**). EMCV-L$^{zn}$-N2aN2b-K$^{257}$A+K$^{261}$A also lost the capacity to suppress the type I IFN response as detected by qRT-PCR, although this was noticeable only at 8 hr post infection (**Fig 7E**). This may be attributed to the considerable delay in virus replication and a consequential late onset of IFN induction.

We conclude that SARS-CoV-2 N2b functions as an antagonist of the ISR in EMCV-infected cells and can functionally replace the EMCV L protein by acting as a classical PKR

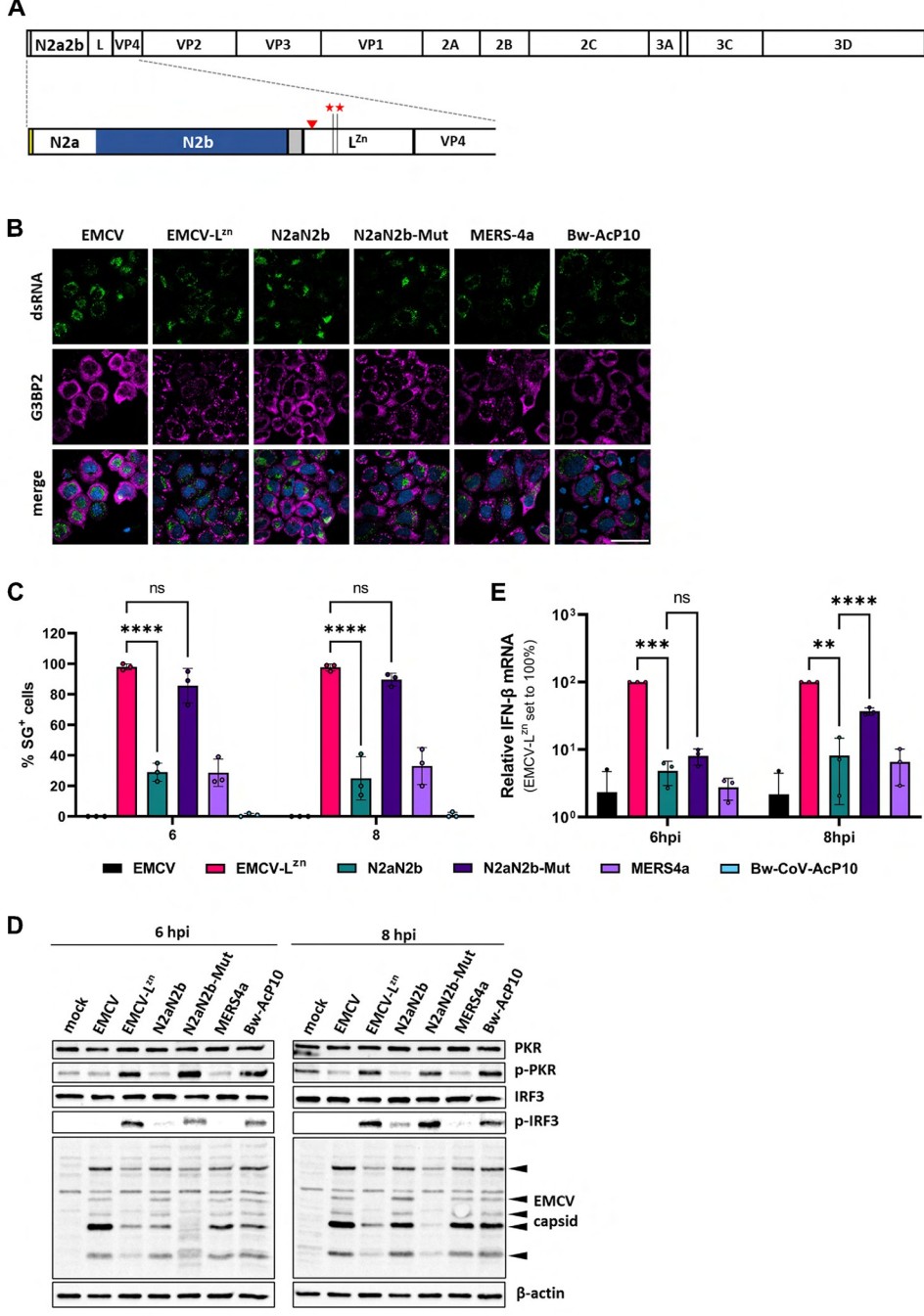

**Fig 7. The N2b domain inhibits PKR phosphorylation, prevents SG formation, and reduces the type I interferon response in EMCV infected cells.** (*A*) Structure of the polyprotein of recombinant EMCV-L$^{zn}$-N2aN2b. Schematic representation, mature cleavage products shown as boxes. Red asterisks indicate the locations of mutations (C$^{19}$A/ C$^{22}$A) in the Zn-finger domain of leader protein L, the red arrowhead that of a newly engineered EMCV 3C cleavage site and the yellow and grey boxes the native N-terminus of the EMCV polyprotein (see **S8 Fig** for details) and a Strep2 tag, respectively. (*B*) SG formation in EMCV-infected cells is prevented by SARS-CoV-2 N2a2b. HeLa cells were infected at MOI 10 with EMCV-L$^{zn}$-N2aN2b$^{wt}$ or EMCV-L$^{zn}$-N2aN2b-K$^{257}$A+K$^{261}$A ('N2a2b-Mut'). Cells infected with wildtype EMCV or EMCV-L$^{zn}$ served as controls as were cells infected with recombinant EMCV-L$^{zn}$ derivatives encoding established coronavirus ISR antagonists MERS 4a and BW-Acp10. Cells fixed at 8 hpi were analyzed by IFA for dsRNA as a marker for infection and for SG scaffold protein G3BP2. (*C*) Percentages of infected cells with stress granules at 6 and 8 hpi. Bar graphs show the means of three independent experiments with >200 infected cells counted per sample. Standard deviations indicated by error bars; **** p< 0.0001; ns, non-significant (two-way ANOVA with

Dunnett post hoc test). (**D**) Western-blot analysis for PKR, phosphorylated PKR (p-PKR), IRF3, phosphorylated IRF3 (p-IRF3), EMCV capsid proteins and β-actin. HeLa cells, (mock-)infected with recombinant EMCVs at MOI 10, were lysed at 6 or 8 hpi. Images are representative of three independent experiments. (**E**) Suppression of type I interferon response in EMCV-infected cells by N2a2b but not N2aN2b-Mut. Infected HeLa cells (MOI 10) were lysed at 6 or 8 hpi. Total RNA was analyzed by RT-qPCR for IFN-β and actin. IFN-β levels were calculated as fold induction compared to levels in mock-infected cells, after correction for actin mRNA levels, and normalized with EMCV-L$^{zn}$ IFN-β levels set at 100%. Data represent means of three independent experiments. Standard deviations indicated by error bars; statistical significance compared to the results for EMCV-L$^{zn}$ or EMCV-L$^{zn}$-N2aN2b infected cells calculated by two-way ANOVA with the Dunnett post hoc test; ** $p<0.01$; *** $p<0.001$; **** $p<0.0001$.

antagonist and preventing activation of the ISR by sequestering dsRNA. This mode of action differs from that of beluga whale CoV AcP10 and Aichivirus L, which counteract the ISR by salvaging global translation through competitive inhibition of p-eIF2–eIF2B association [31]. The data lend support to the notion that N through N2b may inhibit the ISR and SG formation also in the context of SARS-CoV-2-infected cells.

## Discussion

To successfully establish infection, viruses must subdue the intracellular host defense long enough to complete their replication cycle and vigorously enough to curb intercellular immune signaling. The detection of dsRNA, an inevitable byproduct of both RNA and DNA virus replication is central to activation of the innate antiviral defense. The activation of RLRs must be averted that would otherwise culminate, through intra- and intercellular signaling, in the production of type I interferons and other proinflammatory cytokines. Also, while host cell protein synthesis should best be inhibited and kept at a minimum, efficient unperturbed production of viral proteins must be ensured by preventing translational suppression due to activation of the OAS/RNaseL pathway and PKR-dependent induction of the ISR. Recent publications implicated SARS-CoV-2 N protein as an inhibitor of SG formation and an RLR antagonist [33–36,68,69]. Here we confirmed and extended these observations using a well-established plasmid-based expression system [27,31] to induce PKR-mediated ISR activation and to assess inhibition thereof by EGFP-tagged ISR antagonists. We conclusively demonstrate that SARS-CoV-2 N is endowed with the ability to inhibit PKR activation, translational arrest and ensuing SG formation. Moreover, we showed that the N protein of human alphacoronavirus 229E (subgenus *Duvinacovirus*) can also prevent translational arrest and suppresses SG formation to a similar extent and via a similar mode of action. These findings suggest that this property may be widely conserved among members of different orthocoronavirus (sub)genera.

To study the mechanism of ISR inhibition, we performed a systematic deletion and site-directed mutational analysis of SARS-CoV-N. We found that inhibition of SG formation was lost upon deletion of N2b in accordance with findings of Zheng et *al*. [35]. However, we crucially extend their observations by showing that N2b when expressed in isolation, is sufficient to impair transfection-induced ISR-mediated translational shut off as well as the formation of SGs. Apparently, N2b counteracts the ISR by binding dsRNA. Ala substitution of conserved Lys residues (Lys$^{257}$ and/or Lys$^{261}$) in the putative RNA-binding cleft of N2b that abolished dsRNA binding also abolished ISR inhibition. Importantly, these substitutions also caused the intact N proteins of SARS-CoV-2 and HCoV-229E to lose their activity as ISR antagonists. The observations thus strongly support a model in which coronavirus N proteins through their N2b domain prevent activation of the ISR by PKR by sequestering dsRNA, like the other viral PKR inhibitors IAV ns1 and MERS-CoV ns4a that were used as controls throughout.

The N protein plays a crucial role during the very early steps of the CoV infection cycle [56,70–72] and is indispensable for virus morphogenesis. Evidently, this impedes a direct

mutational reverse genetics approach to study the occurrence and relevance of N-mediated inhibition of the ISR in CoV-infected cells. However, by using a mutant EMCV as a platform, we showed that an extended protein N2aN2b can functionally replace the autologous enterovirus L protein and can salvage the EMCV replication defect caused by L inactivation, like we previously showed for MERS-CoV ns4a and BWCoV AcP10 [27,31]. Like MERS-CoV ns4a, but unlike BWCoV AcP10 -which blocks the ISR downstream of PKR-, N2aN2b prevented PKR phosphorylation apparently via N2b-mediated RNA binding. The findings provide proof of principle that N2b can promote viral replication by preventing PKR and ensuing ISR activation, supporting the notion that N through its N2b domain may do so as well during CoV infection.

Adding relevance to our observations, selection for enhanced innate immune escape apparently favored SARS-CoV-2 variants with increased N expression [73]. Moreover, a three-nucleotide change in Gamma also present in Alpha and Omicron variants of concern created a new (cryptic) transcription regulating sequence to drive the synthesis of a novel subgenomic mRNA species from which a truncated N-protein, N*, may be translated corresponding with the C-terminus of the protein and initiating at in frame Met$^{210}$ [73,74]. If N* is indeed expressed, our findings strongly support a role in dsRNA sequestration and innate antagonism.

The native coronavirus N proteins are subject to extensive posttranslational modifications in domains and segments other than N2b. Some have been implicated in liquid-liquid phase separation [63,75], RNA binding [33], SG formation and inhibition of innate immunity [33,63], prompting the question whether these modifications promote or hinder its function as an ISR antagonist. Reportedly, methylation of Arg$^{95}$ in the SARS-CoV-2 N1b segment by host protein arginine methyltransferases is required for RNA binding and for the inhibition of arsenite-induced formation of G3BP1-containing SGs [33]. In our hands, however, an N mutant with Arg$^{95}$Lys substitution, described to block N-mediated suppression of SGs [33], still prevented PKR-induced ISR-associated translational repression to identical extent as the wildtype protein and likewise prevented SG formation. In another recent study, Wang et al. [63] reported suppression of MAVS signaling by intact SARS-CoV-2 N protein but not by an N derivative from which N2b had been deleted. They attributed this to N-induced and N2b-dependent liquid-liquid phase separation (LLPS) subject to acetylation of N3 residue Lys$^{375}$. In accordance, we find that in EMCV-infected cells, N2aN2b, like MERS4a, suppresses β-interferon (IFN-β) expression and show that this activity is blocked by mutations that abolish N2b-mediated RNA binding. In apparent contrast to the observations of Wang et al. [63], however, we find that acetylation of Lys$^{375}$ is not essential for RNA binding by N2b and that Lys$^{375}$Arg substitution does not detectably affect inhibition of the PKR-induced ISR by the intact N protein. Our findings suggest that N, through the autonomous activity of the N2b domain, acts not only as a classical PKR inhibitor to thwart ISR activation but may also prevent RLR activation and thus block induction of the type I IFN response, whether directly by shielding dsRNA or indirectly by preventing SGs to act as platform for immune signaling. The interaction between N and SG scaffold proteins G3BP1 and -2 mediated through the N1a ΦxFG motif [33,37–40], was recently concluded to be the main determinant in SG disassembly [38]. Our current findings would seem to be at odds with this view, but in accordance with observations by these authors, we did observe that functional disruption of the critical ΦxFG motif abrogated inhibition of SG formation in arsenite-treated HeLa PKR$^{KO}$ cells. Our findings, however, would suggest that sequestration of the G3BPs to prevent assembly of SG or to promote their disassembly requires high expression levels of N. Importantly, N proteins with intact N1a domain but with mutations in N2b that abolish RNA binding failed to prevent SG formation in HeLa wt cells. Also, the 229E N protein lacks a ΦxFG motif, yet inhibits formation of SGs

via the same N2b-dependent mechanism as SARS-CoV-2 N. The findings may be reconciled, however, by assuming that sarbecovirus N proteins have been evolutionary selected to interfere with G3BP function, possibly even beyond the formation and function of SGs [76], through distinct mechanisms at multiple levels during different stages of the viral life cycle, mediated by different domains and regulated by distinct posttranslational modifications, and perhaps subject to protein distribution and availability of viral genomes for encapsidation. Such subtleties may be missed in over-expression experiments and overshadowed by the robust inhibition of the PKR-triggered ISR by N2b. Hence, in coronavirus-infected cells, N1a-mediated G3BP-N association [38,40,55] and N2b-mediated dsRNA binding [36] may well act in concert to hamper SG assembly and to impede recruitment and SG-facilitated activation of RLRs and PKR [20–23]. Indeed, this would be consistent with observations by others [35,36] of SARS-CoV-2 N suppressing SG-associated RLR activation and inhibiting induction of type I IFNs by targeting G3BP1. The relative importance of dsRNA shielding, G3BP recruitment and other activities of N for suppression of innate immunity during coronavirus infection clearly deserves further study.

## Materials and methods

### Cell lines

HeLa-R19, HeLa R19 PKR$^{KO}$ [27], A549, BHK21 and Vero E6 cells were maintained in Dulbecco's Modified Eagle's Medium (DMEM) supplemented with 10% (V/V) fetal calf serum (FCS) and 100 units/ml penicillin and streptomycin.

### Recombinant EMCV viruses

Recombinant EMCV viruses were derived from the pM16.1-derived pStrep2-VFETQG-Zn-M16.1 infectious clone [27,77]. This vector carries a Zn-finger mutation to inactivate EMCV L as well as coding sequences for a Strep2-tag and a synthetic optimized 3C$^{pro}$ recognition site (VFETQG). The coding sequences for N2b and N2a2b were PCR amplified from SARS-CoV-2-derived cDNA (sequences deposited in EVA database; Ref-SKU: 026V-03883) and inserted into XhoI/NotI-digested pStrep2-VFETQG-Zn-M16.1 yielding pStrep2-SARS-CoV-2-N2b-VFETQG-Zn-M16.1 and pStrep2-SARS-CoV-2-N2aN2b-VFETQG-Zn-M16.1, respectively. Mutations to substitute N2b Lys$^{257}$ and Lys$^{261}$ by Ala were introduced by Q5 side-directed mutagenesis (New England Biolabs, NEB) generating pStrep2-SARS-CoV-2-N2aN2b K$^{257}$A/K$^{261}$A-VFETQG-Zn-M16.1. The vectors were linearized with BamHI, used for *in vitro* transcription with the RiboMAX kit (Promega) and the resulting RNA was purified using the NucleoSpin RNA mini kit (Machery-Nagel). Viruses were recovered by transfecting BHK-21 cells, grown to subconfluency in T125 flasks, with 1.5 μg of the run-off RNA transcripts using Lipofectamine2000 (Invitrogen). After 2–4 days, when total cytopathic effect was apparent, the cultures were subjected to three freeze-thaw cycles, cell debris was pelleted at 4,000x*g* for 15 minutes and virus was concentrated from the supernatants by ultracentrifugation though a 30% sucrose cushion at 140,000x*g* for 16 hrs at 4˚C in a SW32Ti rotor. Virus pellets were resuspended in phosphate-buffer saline (PBS). The viruses were characterized by isolating viral RNA from 150-μL aliquots of the cell culture supernatant with the NucleoSpin RNA Virus kit (Macherey-Nagel) followed by conventional RT-PCR and bidirectional Sanger sequence analysis of the inserted SARS-CoV-2 sequences and flanking regions. Viral titers, determined by endpoint titration and calculated by the Spearman-Kaerber formula, are averages from three independent experiments.

## Plasmids for eukaryotic and prokaryotic expression

Eukaryotic expression plasmids were constructed by cloning PCR-amplified sequences, flanked by NheI and BamHI restriction sites, into NheI/BamHI digested vector pEGFP-N3 (ClonTech) such that the encoded viral proteins are C-terminally fused to enhanced green fluorescent protein (EGFP). pcDNA-RFP was purchased from Addgene.

Procaryotic expression plasmid pGEX2T-Hisx6-N2b encoding N-terminally His-tagged SARS-CoV-2 N2b was created by linearizing vector pGEX2T, amplifying the N2b coding sequences by Q5-PCR, and assembling the fragments by NEBuilder HiFi DNA Assembly (NEB). Mutations (N2b $K^{257}$A, N2b $K^{261}$A and N2b $K^{257}$A+$K^{261}$A) were generated by Q5 site-directed mutagenesis (New England Biolabs, NEB).

All constructs were sequenced to confirm integrity. There were no major differences in transfection efficiency (S9 Fig)

## Immunofluorescence assay

Cells were seeded onto 12 mm glass cover slips in 24-well plates (Corning Costar) at $5x10^4$ cells, grown for 24 hr and transfected with 500 ng total DNA/well using Lipofectamine2000 (Invitrogen). At 24 hr post transfection, cells were either left untreated or treated with 500 μM sodium arsenite ($NaAsO_2$, Riedel-de Haën) diluted in DMEM for 30 min at 37˚C and subsequently fixed in PBS + 3.7% paraformaldehyde (PFA). Vero E6-TMPRSS2 cells were infected with SARS-CoV-2 Wuhan (D614G) or Omicron BA.1 variants at a multiplicity of infection (MOI) of 5 $TCID_{50}$/cell and HeLa-R19 cells were infected with recombinant EMCV viruses at an MOI of 10 $TCID_{50}$/cell and incubated for times indicated in the text. Cells were fixed with paraformaldehyde (3.7% in PBS), incubated with PBS + 0.1% glycine for 10 min, permeabilized with 0.1% Triton X-100/PBS+s for 10 min at RT and blocked in blocking buffer (PBS + 1% BSA + 0.1% Tween-20) for 30min in a dark humidified chamber at 37˚C. The cells were then incubated in blocking buffer containing mouse anti-dsRNA (1:1000; English & Scientific Consulting), goat anti-eIF3η (1:200, SantaCruz) and rabbit anti-G3BP2 (1:200; Bethyl Laboratories) for 1h at room temperature (RT). After washing with PBS+0.1%Tween-20, cells were incubated with secondary antibody donkey anti-mouse Cy2 (1:100; The Jackson Laboratory), donkey anti-goat Alexa594 (1:200; Invitrogen), donkey anti-rabbit Alexa647 (1:200; Invitrogen) in block buffer for 1h at RT. The cells were then washed three times with PBS+0.1% Tween-20, once with distilled water and mounted on glass microscope slides in ProLong Diamond Antifade (Invitrogen) mounting medium. Cells were examined by conventional widefield (Olympus) and confocal immunofluorescence microscopy (Nikon A1R) in most cases also in a blinded fashion by a second independent observer.

## Flowcytometry analysis

HeLa cells were seeded in a 12-well cluster ($10^5$ cells/well) and, after overnight incubation, transfected with the indicated plasmids (500 ng well; 250 ng/plasmid) using Lipofectamine 2000 (Invitrogen). At 24 hr post transfection, cells were either left untreated or treated with 500 μM sodium arsenite ($NaAsO_2$, Riedel-de Haën) diluted in DMEM for 30 min at 37˚C. Cells were detached with trypsin, washed once PBS, and fixed in PBS, 3.7% PFA for 10 min. To stain for p-eIF2α, cells were permeabilized in ice-cold MeOH for 10min, washed twice in FACS buffer (PBS + 0.02% Na-azide and 1% BSA) and subsequently incubated with the primary antibody rabbit anti-p-eIF2α (1:100, Abcam) in FACS buffer for 45 min. The samples were washed twice in FACS buffer and incubated in the secondary antibody goat anti-rabbit Alexa 594 (1:100, Invitrogen) in FACS buffer for 45 min. Cells were washed twice with FACS buffer and stored at 4˚C. Fluorescence intensity was recorded with a CytoFLEX LX flow

cytometer (Beckman Coulter) and the data were analyzed by FlowJo v10 software (BD Biosciences). Samples were gated for live single cell populations and then gated for negative, low (L) and high (H) RFP expressing cells. For bar graphs [H/L] ratios were normalized with the ratio calculated for the relevant wildtype protein set at 100%.

## Electrophoretic Mobility Shift Assay (EMSA)

To purify recombinant N2b proteins, *E. coli* BL21 cells (Sigma-Aldrich), transformed with pGEX2T-Hisx6-N2b or its mutated derivatives, were grown in LB medium containing ampicillin (50 μg/ml) at 37°C until the optical density at 600 nm ($OD_{600nm}$) reached 0.3. The temperature was then reduced to 18°C, and when the $OD_{600nm}$ reached 0.5, protein expression was induced with 0.5 mM IPTG. Following expression for 16 hrs, the cells were harvested by centrifugation and resuspended in lysis buffer (100 mM Tris-HCl pH = 8, 300 mM NaCl, 10 mM imidazole, 0.1% Triton X-100, 5% glycerol) complemented with lysozyme (0.25 mg/ml; Merk) and cOmplete Protease Inhibitor (Roche). The samples were sonicated and centrifuged at 15,000*g* for 45 min at 4°C to pellet cell debris. The cleared lysates were passed through a 0.45-μm filter and incubated with 1 ml of Ni-NTA (nitrilotriacetic acid) resin (Thermo Scientific) at 4°C overnight on a roller. The beads were washed twice with wash buffer (100 mM Tris-HCl (pH 8.0), 300 mM NaCl, 10mM Imidazole) and then eluted with a buffer consisting of 100 mM Tris-HCl, 300 mM NaCl, and 500 mM imidazole (pH 8). The eluted proteins were subjected to dialysis in 20 mM Tris-HCl, 150mM NaCl and flash-frozen in 20-μl aliquots.

For the EMSA, 1 μM of a 32-mer single stranded RNA oligonucleotides corresponding to SARS-CoV-2 (GenBank accession no. MZ558051.1) sequence 5′-CGAGGCCACGCGGAGU ACGAUCGAGGGUACAG-3′ and a scrambled version thereof, 5′-GGCACGGAGUAUACCGG ACGAGCGGAACGGCU-3′, each were mixed 1: 1 with their respective complimentary RNA oligonucleotides in denaturing buffer (10 mM $NaH_2PO_4 \cdot H_2O$, 50 mM NaCl, 1 mM EDTA, 0.01% $NaN_3$, pH 7.4 at 25°C), incubated at 90°C for 4 min and then allowed to anneal at RT for 30 min. For the ssRNA sample preparation, the oligonucleotide was diluted in denaturing buffer, incubated at 90°C for 30 sec to destroy possible secondary structures and rapidly cooled on ice. Proteins were diluted in protein buffer (20 mM Tris-HCl (pH = 8), 150 mM NaCl) to a concentration of 100 ng/μl and mixed in 10X, 20X or 40X fold molar excess with 10 ng ssRNA or 10 ng dsRNA in binding buffer (20 mM Tris-HCl, 100 mM NaCl, 1 mM EDTA, 1mM TCEP, 0.02% Tween-20, pH 7.0 at 25°C) in 5 μl reaction volumes. The samples were incubated for 30 min on ice, then supplemented with glycerol to a final concentration of 5% (v/v) and separated in ultrathin (10 ml; 75 x 50 mm) RNase-free 1% agarose gels in 0.5x TB buffer (45 mM Tris base, 45 mM boric acid, pH 8.2–8.5) at 200V. RNAs were stained by incubating the gels in 50 ml 2X Invitrogen SYBR Gold nucleic acid gel stain (ThermoFisher) in 0.5x TB buffer, diluted from a 10.000X concentrate, for 15 min under agitation and de-stained for 21 min with 0.5 X TB buffer refreshing the buffer 3 times. Stained RNA was visualized with the Gel Doc System (BioRad).

## Western-blot analysis

HeLa R19 wt and PKR[KO] cells were seeded in 6-well clusters (4x10[5] cells/well) and after a 16 hr recovery were infected with recombinant EMCVs at MOI 10 or transfected with plasmids as indicated in the text. The infection was allowed to proceed for 6 or 8 hrs and transfections for 24 hrs. Cells were then released by trypsin and lysed in ice-cold lysis buffer (50 mM Tris-HCl pH 8.0, 150 mM NaCl, 1 mM EDTA, 1% NP40, cOmplete Mini Protease Inhibitor Cocktail (Roche) and phosphatase inhibitor PhosSTOP (Roche)) for 30 min at 4°C under constant agitation. Cell debris was pelleted for 20 min 12000 rpm at 4°C. Cleared lysates were harvested

and protein concentrations determined by the Pierce BCA assay (ThermoFisher Scientific). Samples of each lysate, corresponding to 100 μg protein, were separated by SDS-PAGE in reducing 10% polyacrylamide gels. For western-blot analysis of eIF2α phosphorylation, cells were lysed with SDS sample buffer (50 mM Tris-HCl pH 6.8, 1 M Glycerol, 1.5 mM Bromophenol Blue, 100 mM DTT, 2% SDS w/v). Lysates were sonicated for 15 sec to shear chromosomal DNA and subsequently analyzed by SDS-PAGE. Lysates from arsenite-treated mock-transfected cells were included as positive control. The proteins were then blotted onto 0.2 μm nitrocellulose membranes by wet electrophoretic transfer or semi-dry transfer. Membranes were washed three times in TBST (20 mM Tris, 150 mM NaCl + 0.1% Tween-20), 5 min each, and incubated in blocking buffer (TBST + 2% BSA) for 1h at RT. Membranes were successively incubated with primary antibodies diluted in blocking buffer (mouse anti-G3BP1, 1:4000, BD Biosciences, rabbit anti-GFP, 1:1000, ThermoFisher, mouse anti-PKR, 1:1000, BD Biosciences; rabbit anti-PKR-P, 1:1000, rabbit anti-eIF2α, 1:1000, Cell Signaling; rabbit anti-eIF2α-P, 1:1000, Abcam; rabbit anti-IRF3, 1:1000, Abcam; rabbit anti-IRF3-P, 1:1000, Cell Signaling; Abcam; mouse anti-βactin, 1:5000, Invitrogen; rabbit anti-mengovirus capsid, 1:1000, kindly provided by Prof. Ann Palmenberg) for 16 hr at 4˚C, and then for 30 min at RT with goat-α-mouse-IRDye680 (Li-COR, 1:15000) or goat-α-rabbit-IRDye800 (Li-COR, 1:15000) diluted in blocking buffer. Between and after the incubations, the membranes were washed, thrice each time, with TBST. Finally, membranes were washed once with PBS and scanned using an Odyssey Imager (Li-COR). ImageJ was used to quantitate and compare density of bands after correction with beta-actin as loading control.

## Co-immunoprecipitation (Co-IP) assay

HeLa R19 wt and PKR$^{KO}$ cells were seeded in 6-well clusters (4x10$^5$ cells/well) and, after a 16 hr recovery, transfected with the indicated plasmids. At 24 hrs post transfection, cells were washed once in PBS, released by trypsin and incubated in ice-cold lysis buffer (Tris-HCl pH 8.0, 50mM, NaCl 150mM, EDTA 1mM, NP40 1%, cOmplete Mini Protease Inhibitor Cocktail (Roche) and phosphatase inhibitor PhosSTOP (Roche)) for 30 min on ice. The cell lysates were cleared in a microcentrifuge for 10 min at 12000 rpm at 4˚C. Supernatants were harvested and incubated with 25 μl of preequilibrated GFP-Trap agarose beads (ChromoTek) for 1h at 4˚C with end-over-end rotation. Beads were then washed 3 times with wash buffer (50 mM Tris-HCl pH 8.0, 150 mM NaCl, 1 mM EDTA). Bound proteins were eluted in 80 μl 2x SDS-sample buffer, separated by SDS-PAGE in reducing 8% polyacrylamide gels and analyzed by Western blotting.

## RT-qPCR analysis

HeLa R19 cells, seeded in 24-well clusters (5x10$^4$ cells/well), were inoculated with recombinant EMCVs at MOI 10 as above. At 6 or 8 hrs post infection cells were lysed and cellular RNA was isolated using the total RNA isolation kit (Machery-Nagel). Reverse transcription was set up using TaqMan Reverse Transcription Reagents (Applied Biosystem). qPCR analysis of human IFN-β, human actin mRNA and EMCV viral RNA was performed mixing the Fast SYBR green Master Mix (ThermoFisher) with cDNA and 1μM of the forward and reverse corresponding primers: IFN-β (5′-ATGACCAACAAGTGTCTCCTCC-3′ and 5′-GCTCATGGAAAGAGC TGTAGTG-3′), actin (5′-CCTTCCTGGGCATGGAGTCCTG-3′ and 5′GGAGCAATGATCT TGATCTTC-3′) and EMCV RNA (5′-TCTGTTCTGCCTGTGTTTG-3′ and 5′-AAAGAAG AGGGTGCCGAAAT-3′). Amplification occurred in a Roche Light Cycler using the following program: polymerase activation (95˚C for 5 min), amplification (45 cycles: 95˚C for 10 sec, 60˚C for 5 sec, 72˚C for 30 sec), melting curve (95˚C for 5 sec, 65˚C for 1 min) and cooling

(40°C for 10 sec). The experiments were carried out in triplex for each data point. The relative quantification of the IFN-β gene expression was determined using the $2^{-\Delta\Delta Ct}$ method [78], then the relative IFN-β mRNA levels were normalized to the EMCV-zn IFN-β mRNA level set as 100.

## Supporting information

**S1 Fig. (Related to Fig 1).** (***A***) **Transfection with pEGFP-based transfection vectors induces SG formation.** Hela wt cells, transfected with pEGFP, were stained for SG markers. Granules positive for G3BP2 also contain G3BP1, TIA-1 and eIF3. EGFP intensity digitally increased with respect to standard conditions to show all EGFP+ cells (see below). Size bar: 50 μm. (***B***) **Cell-to-cell heterogeneity in pEGFP-driven EGFP and N-EGFP expression levels.** Hela wt cells were transfected to express EGFP, SARS-CoV-2 N-EGFP or SARS-CoV-2 N-N2b mut-EGFP, i.e. a derivative unable to inhibit the ISR (*vide infra*). Expression levels were assessed by fluorescence microscopy as in Fig 1A. Images acquired at standard (low) EGFP intensity laser power (top) and high intensity laser power (bottom) to illustrate that all cells containing SGs are in fact transfected. Size bar: 50 μm.
(TIF)

**S2 Fig. SARS-CoV-2 infection does not trigger SG formation in Vero E6 cells.** Vero E6 cells stably expressing TMPRSS2 were infected with SARS-CoV-2 Omicron BA.1 and Wuhan (D614G) variants at MOI 5. After 6, 8 and 16 hpi cells were fixed and stained with antibodies against dsRNA as an infection marker, and eIF3 and G3BP2 as SG markers. Mock-infected cells were treated with sodium arsenite to induce SGs and stained with antibodies against G3BP1, G3BP2 and eIF3. Size bar: 50 μm.
(TIF)

**S3 Fig. (Related to Fig 1). SARS-CoV-2 N and HCoV-229E N proteins prevent eIF2α phosphorylation.** Hela wt cells were (mock)transfected to express EGFP, SARS-CoV-2 N-EGFP, SARS-CoV-2 N-N2b mut-EGFP, HCoV-229E N-EGFP and HCoV-229E N-N2b mut-EGFP. Phosphorylated eIF2α (p-eIF2α) levels were assessed by flow cytometry. The dashed line divides p-eIF2α negative (left) from p-eIF2α positive (right) cell populations. The percentage of positive p-eIF2α cells is indicated in the top right part of each panel.
(TIF)

**S4 Fig. (Related to Fig 1). Basal expression levels of N-EGFP and EGFP.** Hela wt and HeLa PKR$^{KO}$ cells were (mock)transfected to express EGFP, SARS-CoV-2 N-EGFP and N-N2b mut-EGFP, i.e. a derivative unable to inhibit the ISR. (***A***) Expression levels assessed by flow cytometry. Cell counts plotted against EGFP fluorescence intensity. The dashed line marks maximum fluorescence intensity observed in pEGFP-transfected, HeLa wt cells. (***B***) Expression levels assessed by fluorescence microscopy as in Fig 1A. The N-EGFP fusion protein is non-codon optimized and three times larger in size than codon-optimized EGFP. Note that expression of EGFP, as indicated by average fluorescence intensity, is higher than that of N-EGFP when compared in HeLa PKR$^{KO}$ cells, i.e. in the absence of translational arrest. However, also note that in HeLa wt cells, under conditions of PKR-mediated ISR-induced translational arrest, (i) expression of EGFP is restricted and (ii) as a result, expression of N-EGFP greatly exceeds that of EGFP in a sizeable population of transfected cells. The analyses have been performed for all the constructs used in this study; data available upon request. Size bar: 50 μm.
(TIF)

**S5 Fig. (Related to Fig 4). Mutagenesis of the serine rich region (SR) of SARS-CoV-2 N alters the cellular distribution of N but does not affect its ability to inhibit ISR-induced translation arrest.** HeLa R19 cells were transfected to express SARS-CoV-2 N (13S>A)-EGFP. Transfected cells were stained for eIF3 and G3BP2 as markers for SGs. Rather than distributing throughout the cytoplasm, the mutant protein accumulates in large local deposits resembling aggregates. These accumulates contain G3BP2 but in most cases are devoid of eIF3 (cells marked with white crosses). Note that the fluorescence intensity in N (13S>A)-EGFP expressing cells is comparable to that in cells expressing parental N-EGFP (see e.g. **Figs 1** and **4**) and consistently higher than in cells expressing EGFP alone or N-N2b mut-EGFP. Size bar: 50 μm.
(TIF)

**S6 Fig. (Related to Fig 4). N2b-defective N-EGFP derivatives are recruited to SGs but strictly dependent on the presence of a functional G3BP binding site in subdomain N1a.** Hela wt cells were transfected to express SARS-CoV-2 N-N2b mut-EGFP or SARS-CoV-2 N-N1a+N2b mut-EGFP and stained for SGs. Due to the mutations in N2b (K257A+K261A) SGs are formed in both cases, but only N-N2b mut-EGFP co-localizes with SGs. Apparently, the I14A+R15A substitutions in the ΦXFG G3BP1 binding motif in N1a abrogate SG recruitment. EGFP intensity digitally increased with respect to standard conditions to show all EGFP + cells. Size bar: 50 μm.
(TIF)

**S7 Fig. (Related to Fig 6). Co-transfection assay to test for translational repression *in trans*.** HeLa cells were transfected to express EGFP, EGFP-tagged N domain N2b, EGFP-tagged full length N and mutant derivatives thereof from pEGFP-N3-based expression vectors to induce PKR-activated ISR. The capacity of these proteins to rescue translational repression *in trans* of red fluorescent protein (RFP), expressed from vector pcDNA-RFP, or lack thereof was measured by fluorescence microscopy. Scale bar: 50 μm.
(TIF)

**S8 Fig. (Related to Fig 7). N-termini of the rEMVC-L$^{zn}$-N2b and rEMVC-L$^{zn}$-N2a2b polyproteins.** Indicated are the N-terminal six residues of EMVC L, the two residues encoded by an engineered *Xh*oI cleavage site fused to SARS CoV-2 N residues 248–365 or 176–365, respectively.
(TIF)

**S9 Fig. Transfection efficiency of N-derivative constructs in HeL and HeLa PKR$^{KO}$ cells.** Transfection efficacy of transfected constructs used throughout this study has been measured by flow-cytometry in HeLa-R19 wt and HeLa-R19 PKR$^{KO}$ cells.
(TIF)

**S1 Table. (Related to Fig 2A). Comparison of mean percentages of transfected cells with SGs calculated from three biological triplicates, showing the decrease in %, as compared to the control (pEGF-N3-transfected cells, "GFP").** Ordinary One-way ANOVA, Dunnett's multiple comparison test.
(TIF)

**S2 Table. (Related to Fig 3D). Comparison of mean percentages of transfected cells with SGs calculated from three biological triplicates, showing the decrease in % as compared to the control (pEGF-N3-transfected cells, "GFP").** Ordinary One-way ANOVA, Dunnett's multiple comparison test.
(TIF)

**S3 Table. (Related to Fig 5B and 5C). Comparison of mean percentages of transfected cells with SGs calculated from three biological triplicates, showing the decrease in % as compared to wt N or the mutant N1a+N2b N.** Ordinary One-way ANOVA, Dunnett's multiple comparison test.
(TIF)

## Acknowledgments

We would like to thank Huib H. Rabouw for helpful comments and discussions, Richard Wubbolts and Ester van t Veld from the Center for Cell Imaging for the microscopy support, Ger Arkesteijn and Estefania Lozano Andres from the Flow Cytometry and Cell Sorting Facility for the flow cytometry support, Robert Creutznacher for making the structure representations and Jolanda de Groot-Mijnes for critical reading of the manuscript.

## Author Contributions

**Conceptualization:** Chiara Aloise, Jelle G. Schipper, Raoul J. de Groot, Frank J. M. van Kuppeveld.

**Formal analysis:** Chiara Aloise, Jelle G. Schipper, Daniel L. Hurdiss, Raoul J. de Groot, Frank J. M. van Kuppeveld.

**Investigation:** Chiara Aloise, Jelle G. Schipper, Arno van Vliet, Judith Oymans, Tim Donselaar.

**Supervision:** Raoul J. de Groot, Frank J. M. van Kuppeveld.

**Visualization:** Chiara Aloise, Jelle G. Schipper.

**Writing – original draft:** Chiara Aloise, Raoul J. de Groot, Frank J. M. van Kuppeveld.

**Writing – review & editing:** Chiara Aloise, Jelle G. Schipper, Arno van Vliet, Judith Oymans, Tim Donselaar, Daniel L. Hurdiss, Raoul J. de Groot, Frank J. M. van Kuppeveld.

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
