## [Decision Letter · Decision Letter 0]

25 Apr 2023

Dear Frank,

Thank you very much for submitting your manuscript "SARS-CoV-2 nucleocapsid protein inhibits the PKR-mediated integrated stress response through RNA-binding domain N2b" for consideration at PLOS Pathogens. As with all papers reviewed by the journal, your manuscript was reviewed by members of the editorial board and by several independent reviewers. In light of the reviews (below this email), we would like to invite the resubmission of a significantly-revised version that takes into account the reviewers' comments.

The reviewers appreciated the technical soundness of the manuscript and importance of the new results, but suggested some additional experiments requred to fully support the conclusions. 

We cannot make any decision about publication until we have seen the revised manuscript and your response to the reviewers' comments. Your revised manuscript is also likely to be sent to reviewers for further evaluation.

Sincerely,

George A. Belov, PhD

Academic Editor

PLOS Pathogens

Alexander Gorbalenya

Section Editor

PLOS Pathogens

Kasturi Haldar

Editor-in-Chief

PLOS Pathogens

orcid.org/0000-0001-5065-158X

Michael Malim

Editor-in-Chief

PLOS Pathogens

orcid.org/0000-0002-7699-2064

Reviewer's Responses to Questions

**Part I - Summary**

Reviewer #1: Multiple groups have shown that coronavirus N proteins inhibit stress granule (SG) formation, but the precise mechanism remains obscure, with most studies quite focused on physical interactions with G3BP proteins. Here, Aloise, C., et al. conduct a methodical study mapping the determinants of N-mediated SG suppression. They show the N2b (aka CTD) domain that is involved in RNA binding and N dimerization is sufficient for SG suppression whereas other domains were dispensable. Substitution of key lysine residues implicated in RNA binding inactivated the N2b domain (both for SARS-CoV-2 N and hCoV-229E N), whereas substitution of key glutamine and arginine residues linked to dimerization did not affect SG formation. Thus, RNA binding appears to be central to the SG suppression mechanism. By contrast, substitution of key residues involved in N acetylation and arginine methylation showed that these PTMs are not involved in SG suppression. Similarly, the amino-terminal G3BP binding domain was dispensable for PKR-induced SG formation. Finally, citing the challenges in making and rescuing CoVs with mutations in N, the authors engineered a recombinant EMCV where the SG antagonist (L) is replaced by a portion of SARS-CoV-2 N encompassing N2a and N2b. This recombinant virus suppressed SG formation, and mutating the same key lysines involved in RNA binding reversed this phenotype.

Overall, this manuscript comprises a methodical study of the determinants of N-mediated SG suppression, which clarifies some of the unanswered questions and confusion in the field. Experiments are carefully controlled and the data is clearly presented. The data is high quality, and provide the clearest picture to date of this property of N.

Reviewer #2: The paper by Aloise et al. describes a detailed functional assessment of the SARS-CoV-2 N protein and its interference with the ISR. The authors show that binding of N to dsRNA is critical to inhibit the PKR-dsRNA initiation of the ISR.

The manuscript is very well-written, applies solid methodology and highlights the novelty of the SARS-CoV-2 and, in parts, HCoV-229E N-dependent ISR effect in the context of other published articles on SARS-CoV-2 N.

I have several comments mainly related to the dynamics of the SG formation and the specificity of SG detection in the submitted manuscript.

Reviewer #3: This study investigates the mechanisms by which the SARS-Cov-2 nucleocapsid (N) protein inhibits induction of the integrated stress response (ISR) and stress granule (SG) formation. Four previous studies showed that the SARS-CoV-2 N protein inhibited SG formation through RNA binding and interaction with G3BP. The authors first show that the expression of EGFP from a pEGFP-N3 plasmid is low and that SGs are induced. When EGFP is fused to a viral protein that is an ISR inhibitor, the expression of EGFP is much more efficient and SGs do not form. The authors previously used this protocol to analyze ISR inhibition by the AcP10 and 4a proteins of other coronaviruses. In the current study, SARS-CoV-2 N fused to EGFP enhanced translation as efficiently as proteins from 3 other viruses known to inhibit the ISR. Also, EGFP expression was enhanced in PKR KO cells in the absence of an N fusion protein indicating that the decreased expression of EGFP from the plasmid is PKR mediated. The N proteins of both SARS-CoV-2 and the 229E coronavirus inhibited PKR phosphorylation. However, Western blot data confirming N mediated inhibition of eIF2α phosphorylation were not included. Individual N protein domains were next tested for their involvement in inhibition of the ISR. Only the 2nd ordered domain (N2b) was found to be required for ISR inhibition. The two ordered domains were previously suggested to be RNA binding domains involved in nucleocapsid assembly. N also has a known N-terminal G3BP binding site and an SR region that can be phosphorylated. Based on the crystal structure of an N2b dimer, mutations in the proposed RNA binding cleft were designed and the mutant N2bs were tested for their ability to inhibit the ISR. N2b domains with single and double AA mutations in residues proposed to be involved in RNA binding inhibited RNA binding and did not suppress the ISR. Neither mutation of 13 or the 14 SR phosphorylation sites or the acetylation site affected the ability of N to suppress the ISR. Mutation of 1 to 4 residues in the G3BP binding site also did not affect the ability of N to inhibit SG formation in the context of the PKR signaling pathway. However, the effect of these mutations on the N-G3BP interaction was not analyzed. The involvement of an N-G3BP interaction in inhibition was next analyzed in PKR KO cells with SGs induced by arsenite treatment. Some reduction in the % of cells with SGs was observed when the G3BP binding site was mutated suggesting that the interaction of the N1a region with G3BP might have a partial role on reducing SG formation. Flow cytometry assay of dual fluorescence reporters was used to confirm that the N2b domain is mainly responsible for the suppression of the PKR-induced ISR. Finally, the ability of SARS-CoV-2 N2b expressed from the 5′end of the EMCV polyprotein to suppress SG formation was investigated and it was found to be almost as efficient as the EMCV L protein.

The SARS-CoV-2 N protein has many functions and the involvement of each of the known functional domains in suppression of the ISR and SG formation was investigated. Previous studies showed that the SARS-CoV-2 N protein was an inhibitor of SG formation and an RNA sensor antagonist. The present study confirmed and extended the previous observations using multiple techniques and also showed that the N protein of 229E had the same mode of action. The contributions of each of the N domains was tested. The data obtained adequately support for the authors’ conclusion that the N protein N2b domain binding to dsRNA and preventing PKR activation and SG formation is the primary mechanism involved. The authors’ data did not support previous data suggesting that acetylation of Lys 175 is required for N RNA binding or that substitution of this residue reduces N suppression of the ISR. The authors’ data also did not support previous data suggesting that the N-G3BP interaction was the main mode of SG inhibition by N. However, the authors’ conclusions about N mediated reduced induction of Type I IFN and statements about suppression of cellular RNA sensor activation by N2b binding to dsRNA were not supported by any data included in the manuscript.

**Part II – Major Issues: Key Experiments Required for Acceptance**

Reviewer #1: The use of recombinant EMCV expressing a portion of N as a tool to investigate SG suppression during infection is a clever approach. Others have expressed N in trans to complement a N-deficient SARS-CoV-2 (Leo L, et al. Science Bulletin, 2021) and many reverse genetic systems have been developed to generate mutant CoV genomes with precise mutations (BACmid, TAR, CPER, etc.). It is undeniable that N has many important functions in CoV infection and the virus cannot replicate without it, but substituting one key lysine in N involved in SG suppression is a fairly conservative approach that could help us understand the threat that SGs pose to CoVs and corroborate findings from the ectopic overexpression studies. This is a valuable experiment that is well within reach. If a SARS-CoV-2 or hCoV-229E virus with a K257A substitution in N causes SG formation during infection and fails to replicate, then this considerably strengthens the case of these investigators.

Reviewer #2: 1. SG formation is a very complex and dynamic process and can also be transient during virus infections. In this regard I have several remarks:

1a) Although the applied plasmid-based SG overexpression system has been published before, I would suggest to provide more information on the SG dynamics. Did the authors try different time points with and without different stressors/ different nucleic acid stimuli including SARS-CoV-2 RNA/ virus infections to identify the optimal timing and conditions for SG formation and detection?

1b) The source of the stress (pEGFP-N3 transfection) used to induce SG formation seems quite artificial to me. The authors should explain in more detail why the transfection of this plasmid is enough to induce the SG formation and how this reflects the “natural” infection-induced SG formation. The citations do not go back to a primary source showing this phenomenon as far as I can tell. They could cite for example https://journals.plos.org/plosone/article?id=10.1371/journal.pone.0043283

1c) There are several instances where SGs appear in seemingly untransfected cells e.g. Fig 1B EGFP, Fig 2B EGFP, 366-419, Fig 3 EGFP, K257A, Fig 4A EGFP, N1a + N2b mut, Fig 5 EGFP, 1A mut, 2A mut, 3A mut. Is this regarded as background noise or might there be unwanted stimulations e.g. by endotoxins in the plasmid preps?

1d) In Fig 1B the authors show a striking difference in the GFP levels between cells transfected with pEGFP-N3 and several viral genes/orfs that are expressed as fusion proteins with EGFP-N3. SARS-CoV-2 N, MERS-4a and IAV-NS1 increase the GFP levels immensely, which the authors show to be due to inhibition of the integrated stress response (shown by lack of SG formation and absence of PKR phosphorylation). As a control they also transfect PKR knockout cells in which the difference is abrogated. In a GFP Western blot they show unexpected protein level differences between the ISR-antagonizing proteins, which they regard as an artefact possibly caused by differences in baseline transcription due to lack of codon optimization of the fused protein. They do not address, however, the discrepancy of protein levels between WB and IFA in the HeLa wt cells, where GFP-N and GFP-NS1 show much higher fluorescence but seemingly lower protein levels. Please mention in more detail how this can be explained.

2. The use of a single SG marker (anti-G3BP2 antibody) is in my opinion not sufficient. The authors should clarify how specific the staining is by comparing it with other SG markers (e.g. TIA-1). This is particularly important as G3BP proteins can directly interact with the nucleocapsid proteins.

Reviewer #3: The effect of N mutations on the N-G3BP interaction was not analyzed.

Although the title of the final Results section indicates that the effect of N2b on the Type 1 IFN response was also tested, no Fig.7 E panel showing these data were included. Also, no data confirming the statement that this effect was due to reduced activation of RIG-I were included.

**Part III – Minor Issues: Editorial and Data Presentation Modifications**

Reviewer #1: Additional comments:

Figure 3A. The description of N structure in the text is quite informative. Can the cartoon model be improved to match this description to help the reader appreciate these details?

Figure 3D. Testing the Q272A+Q289A mutants and the R276A+R293A mutants was a good idea to more tightly link the phenotype to RNA binding, but since it is stated that these residues are involved in the dimer interface, it should be feasible to demonstrate that dimers do not form properly when these substitutions are made. The citation alone is not sufficient.

There is a lot of discussion of how interfering with SG formation influenced the evolution and expression levels of sarbecovirus N proteins, but this is quite speculative.

Reviewer #2: 1. In some figures only SE-bars are shown, while in some there are data points. I think it would be preferable to show data points in all Figures.

2. Representing number of SG per cell rather than SG-positive cells would allow a better quantitative analysis of the implication of N2b domain in SG formation. For example, the paper by Kruse et al. detected intensity of G3BP1 foci instead of % positive cells. Using this as a readout might fully replicate their results in the context of arsenite induced SGs (Fig 5).

3. In the labeling of the EGFP-viral protein fusion constructs it might be helpful to not only write the fused protein but make clear that its fused to GFP (e.g. instead of SARS-CoV-2 N, EGFP - SARS-CoV-2 N)

4. It would add to the argument to not only see GFP and SG levels as an ISR readout but also p-PKR levels in Fig 2 and 3

Reviewer #3: L123-125 - “Transfection of cells with specific expression plasmids like pEGFP-N3 triggers the ISR through dsRNA-mediated PKR activation, resulting in translation arrest and the formation of SGs.” As written, it suggests that transfection of DNA induces the ISR. Why this plasmid induces the ISR should be explained. Is it due to RNA stem loops in the mRNA NCRs present in this plasmid which would be present as part of the mRNAs transcribed for any inserted coding region or to stem loops in the EGFP coding region?

L364- “transfection of ISR-inducing plasmid DNA” same concern

S Fig 1- The weak signal of the G3BP1 antibody should be acknowledged and used to justify why the rest of the figures have only panels with G3BP2 antibody. A G3BP-1 antibody apparently worked well for IP.

S Fig. 2A, Fig 3 etc - EGFP should be used consistently for figure labels not GFP.

S Fig. 2A- “Cell counts” should be added as the label to the Y axis. The vertical dotted line should be explained in the legend.

S Fig. 2- legend- fuorescence is misspelled (ii) “in result” should be changed to “as a result”

S Fig. 3- The EGFP-N fusion protein is what is in the foci in some cells. Legend- …show G3BP2 aggregates that are not positive …

L198-201- The meaning of this sentence is not clear as written and needs to be rewritten.

Changing all 13 serine residues to alanine could have a major effect of the protein structure but this possibility is not addressed.

L239-240- There are EGFP fusion protein aggregates but they do no seem to be large and also seem to colocalize with G3BP2 (G3BP1 was not tested) but not eIF3 in many transfected cells. However, the eIF3 staining intensity is low and the large white arrows in SFig 3 often cover areas where the foci are located. There does appear to be colocalization of eIF3 in some of the foci.

L240-241- No evidence is presented to support the statement that the fusion protein/G3BP2 aggregates differ in size, appearance from SGs.

L241- …the EGFP expression levels of the ….

L243- …than those of EGFP alone.

L258- …G3PB1 to a similar extent…

L260- …arrest, the expression levels of these mutants were…

L277- “concentration dependent” It is not clear what data supports this and what is differing in concentration. There was no difference detected in the effects of the 4 mutants tested.

L340- …delayed to a similar …

L354- “raising the alarm” this is “slang” and too vague

PLOS authors have the option to publish the peer review history of their article (what does this mean?). If published, this will include your full peer review and any attached files.

Reviewer #1: No

Reviewer #2: No

Reviewer #3: No
---

## [Decision Letter · Decision Letter 1]

27 Jul 2023

Dear Prof. Dr. van Kuppeveld,

We are pleased to inform you that your manuscript 'SARS-CoV-2 nucleocapsid protein inhibits the PKR-mediated integrated stress response through RNA-binding domain N2b' has been provisionally accepted for publication in PLOS Pathogens.

Best regards,

George A. Belov, PhD

Academic Editor

PLOS Pathogens

Alexander Gorbalenya

Section Editor

PLOS Pathogens

Kasturi Haldar

Editor-in-Chief

PLOS Pathogens

orcid.org/0000-0001-5065-158X

Michael Malim

Editor-in-Chief

PLOS Pathogens

orcid.org/0000-0002-7699-2064

Reviewer Comments (if any, and for reference):

Reviewer's Responses to Questions

**Part I - Summary**

Reviewer #1: The authors have revised the manuscript appropriately in response to critiques from all three peer reviewers. I have no further concerns.

Reviewer #2: The authors addressed all remarks appropriately and I congratulate on this nice study. I have no further comments.

Reviewer #3: The modifications made by the authors adequately address this reviewer's concerns.

**Part II – Major Issues: Key Experiments Required for Acceptance**

Reviewer #1: (No Response)

Reviewer #2: I have no further comments.

Reviewer #3: (No Response)

**Part III – Minor Issues: Editorial and Data Presentation Modifications**

Reviewer #1: (No Response)

Reviewer #2: I have no further comments.

Reviewer #3: (No Response)

PLOS authors have the option to publish the peer review history of their article (what does this mean?). If published, this will include your full peer review and any attached files.

Reviewer #1: No

Reviewer #2: No

Reviewer #3: No

---

## [Editor Report · Acceptance letter]

18 Aug 2023

Dear Prof. Dr. van Kuppeveld,

We are delighted to inform you that your manuscript, "SARS-CoV-2 nucleocapsid protein inhibits the PKR-mediated integrated stress response through RNA-binding domain N2b," has been formally accepted for publication in PLOS Pathogens.

Best regards,

Kasturi Haldar

Editor-in-Chief

PLOS Pathogens

orcid.org/0000-0001-5065-158X

Michael Malim

Editor-in-Chief

PLOS Pathogens

orcid.org/0000-0002-7699-2064